# Invasive and Non-Invasive Analyses of Ochre and Iron-Based Pigment Raw Materials: A Methodological Perspective

## Laure Dayet

UMR5608 Travaux et Recherches Archéologiques sur les Cultures, les Espaces et les Sociétés,
Maison de la Recherche, CNRS-Université Toulouse Jean Jaurès, 31058 Toulouse CEDEX 9, France;
laure.dayet@gmail.com

**Abstract:** Naturally occurring and deeply coloured iron-bearing materials were exploited very early on by human populations. The characterization of these materials has proven useful for addressing several archaeological issues, such as the study of technical behaviors, group mobility, and the reconstruction of cultural dynamics. However, this work poses some critical methodological questions. In this paper, we will review ochre studies by focusing on the analytical methods employed, the limits of non-invasive methods, as well as examples of some quality research addressing specific issues (raw material selection and provenience, heat treatment). We will then present a methodological approach that aims to identify the instrumental limits and the post-depositional alterations that significantly impact the results of the non-invasive analysis of cohesive ochre fragments from Diepkloof rock Shelter, South Africa. We used ochre materials recuperated in both archaeological and geological contexts, and we compared non-invasive surface analyses by XRD, scanning electron microscopy coupled with dispersive X-ray spectrometry (SEM-EDXS), and particle-induced X-ray emission (PIXE) with invasive analysis of powder pellets and sections from the same samples. We conclude that non-invasive SEM-EDXS and PIXE analyses provide non-representative results when the number of measurements is too low and that post-depositional alterations cause significant changes in the mineralogical and major element composition at the surface of archaeological pieces. Such biases, now identified, must be taken into account in future studies in order to propose a rigorous framework for developing archaeological inferences.

**Keywords:** ochre; iron-bearing pigments; characterization; provenance; heat treatment; non-invasive methods; post-depositional alterations

## 1. Introduction

The earliest uses of coloring minerals date to the Paleolithic during the initial development of *Homo sapiens* on the African continent [1–5]. During these early periods, more than 200,000 years ago, evidence for the use of pigments comes in the form of red and yellow geomaterials, such as rock fragments, that have use-wear traces on their surfaces and were recovered in archaeological deposits. Since these initial periods, deeply coloured iron-bearing rocks were almost continuously used by human populations. They are the primary pigments encountered in rock art, from the Paleolithic to historic periods, and have been used in various contexts, from the decoration of multiple buildings and statues during antiquity and the Middle Ages, to the decoration of the human body, clothes, and traditional artifacts, as has been widely documented in ethnographic reports at a global scale. The success of red to yellow iron-based geomaterials as pigments and for a wide range of other, less documented yet no less important, functions—used for its abrasive or siccative properties, used for protection from UV rays, mosquitoes, or even as a medicinal ingredient, etc. [6–14]—may be related to two characteristics of these minerals. Firstly, iron-bearing deposits are very widespread over the surface of the earth, and secondly, their primary components, iron oxides and oxy-hydroxydes such as goethite, a mineral that produces a brown yellow color, or hematite, whose powder is usually characterized by a

deep red color, have strong staining and coating properties [15]. These are often studied, in both archeology and ethnography, as a single category of material under the broad denomination of "ochre", although natural hematite and goethite are considered separately in certain contexts [12,16–21]. Much as is done with other mineral resources exploited by humans, their characterization allows for the reconstruction of past technical and symbolic behaviors, of group mobility and social organization, and more generally of cultural dynamics. Broaching these subjects requires bringing to light key information regarding the operational sequences (chaîne opératoire) that bring into play the choices made during raw material acquisition, transformation, and use [2,19,21–33]. The characterization of ochre materials has become an essential part of archeological pigment studies over the last decades, yet such studies are not without their own methodological issues.

Within the fields of archeometry and the archeological sciences more generally, a wide variety of analytical methods and instruments are used to determine both the composition of ochre and its physico-chemical properties. The study of ochre materials from different archeological sites and time periods has revealed that this category is in fact quite heterogeneous, and that it consists in reality of a significant diversity of geomaterials [2,16,17,19,21,24,26,34–39]. They come from various geological formations with different genesis processes and in most cases these formations were submitted to long histories of diagenesis, epigenesis, alteration, and weathering. Thus, ochre materials are composed of various assemblages of minerals and contain an amount of iron oxide or oxy-hydroxyde varying from 5–10% to upwards of 90% (in mass oxide). Given the considerable variability that has been documented within this large family of coloring materials, their study requires a dedicated methodology and specific care when broaching their geological origin, provenience, and raw material selection on one hand, and their transformation on the other. Given the heterogeneous nature and irregular shape of ochre materials, both in situ and laboratory analyses may present considerable limits that outweigh the main advantages of non-invasive routine procedures acknowledged in the characterization of industrial materials. In fact, methodological questions arise each time one is confronted with a novel archaeological collection of ochre materials or ochre pigments of any form, despite the many years that these materials have been the focus of dedicated methodological research. In the proceeding sections of this paper, we will review the advantages and disadvantages of the current methods used in ochre characterization as well as the methodological choices that have been made in response to sampling constraints and in function of the archeological question addressed. This discussion will be followed by an evaluation of the accuracy and relevance of non-invasive methods via a comparison of analyses conducted on surfaces of cohesive samples (entire rock pieces whose structure is preserved), powder samples, and polished sections. We will characterize both the instrumental limits and effects of alternations related to post-depositional processes.

## 2. A Broad Methodological Review

### 2.1. Choosing the Dedicated Methods: An Introduction

When choosing analytical methods for the characterization of ochre materials, several research parameters must be considered:

- What is the archeological question?
- What is the amount of material, the form of the sample (a piece of rock, residues on a bead, a thin pictorial layer, etc.)?
- Is destructive sampling possible, to which extent?
- Is it a short-term or long-term project?
- What are the facilities I have easy access too?

The two last questions might appear a bit pragmatic, but they are part of the process. They may explain the fact that different choices are made by different authors despite similar research questions and sampling constraints. For this reason, the aim of this review is not to propose a unique methodology but to share some of the most successful solutions that have been developed for ochre characterization in the published literature. Because

of the limited aspect of this paper, we decided to mostly focus on studies of archeological ochre pieces and sediments.

Ochre characterization starts with macroscopic examinations as any other petrologic or mineralogical analyses [21,23,26,30,31,33,40]. The wide range of minerals they are composed of as well as the fine-grained texture they feature as a whole or as a part (their matrix or their cement) constitute limits that are hardly overtaken by simple necked eye or binocular microscope examination [24,28,33,35,37]. In order to take into account their geological and mineralogical diversity, multi-analytical approaches were favoured (Table 1). A common attribute of successful ochre studies is the combination of two main types of analyses: petrographic or mineralogical analyses on one hand; elementary analyses on the other (see e.g., [4,24,25,28,29,32,34,40–47]).

The mineralogical composition of ochre materials can be determined by X-ray diffraction (XRD), Raman spectrometry (RS), Fourrier transformed infrared spectrometry (FTIR), petrography, and less frequently by thermogravimetry or Mössbauer spectrometry. X-ray powder diffraction (XRPD) is invasive but it is global, very accurate thanks to exhaustive international reference database such as the PDF (powder diffraction files) from the ICDD (International Centre for Diffraction Data ®), and also semi-quantitative. Petrography is another destructive method, it requires the preparation of polished thin sections, but its advantage is its high precision in the determination of the arrangement of the minerals and their crystalline nature [16,35,37,42,48,49]. Micro-Raman spectrometry is a point by point method, allowing the analyses of a very small volume, single grains for instance, which makes it ideal for the analyses of ochre powder deposits and coatings, including drawings and paintings [42,50–55]. Scanning electron microscopy coupled with dispersive X-ray spectrometry (SEM-EDXS) allows global elemental analyses but it is also a key method for the identification of mineral phases at the surface of a sample of any form, with or without preparation (residues on artifacts, pictorial layer, cohesive rock, thin section, etc.) thanks to its imagery in back scattered electron highlighting chemical contrasts in combination with X-ray analyses or X-ray cartography (EDXS). If there was one method that could be recommended, this would be the SEM-EDXS. As in any other fields of mineralogy, the combination of several of methods greatly improves the accuracy in the process of identifying mineralogical phases and reduces the risks of confusions between minerals with very broadly similar signals.

In parallel, the whole range of elemental analyses used in geochemistry has been used to characterize ochre materials [17–19,36,38,39,56–65]. The main methods encountered are X-ray fluorescence (XRF, portable or in laboratory), PIXE (particle-induced X-ray emission), inductively coupled plasma and atomic emission spectroscopy (ICP-AES), neutron activation (NAA), or inductively coupled mass spectrometry (ICP-MS, with laser ablation, LA-ICP-MS). They are used to identify major and trace elements in ochre materials. The sensitivity of each instrument is different, NAA and ICP-MS being generally considered as the most sensitive methods for trace element quantification. Detection limits also vary accordingly to the conditions of analyses (see e.g., [39]).

**Table 1.** Non-exhaustive list and short description of studies dedicated to the characterization of archeological ochre materials (mainly cohesive pieces and sediments) during the last two decades.

| Reference | Archaeolgical Issue | Type of Remains | Sampling Limits | Method | Procedure | Sample Preparation |
|---|---|---|---|---|---|---|
| Attard Montalto et al. [60] | Provenance (provenience) | Ochre powders and pieces | None | ICP-AES | Invasive | Grinding, dissolution |
| Barham [2] (Young [66]) | Evidencing human exploitation | Ochre pieces | None | XRD<br>ICP-MS<br>SEM-EDXS | Invasive?<br>Invasive<br>Invasive | ?<br>Dissolution<br>Polished blocks |
| Beck et al. [67] | Classification, provenance (provenience) | Ochre pieces | Use-wear traces on the pieces | PIXE | Non-invasive | None |
| Belli et al. [68]; Gialanela et al. [43] | Heat treatment | Ochre lumps | None | XRD<br><br>SEM-EDXS<br>TEM<br>Raman spectrometry | Invasive<br><br>Invasive<br>Invasive<br>Invasive | Removing of the external part, gentle grinding<br>Same<br>Same, suspension, one drop<br>Same |
| Bernatchez [69] | Classification | Ochre pieces | None | XRPD<br>PIXE | Invasive<br>Invasive | Grinding, powder<br>Grinding, pellets |
| Brooks et al. [1] | Evidencing human exploitation, provenance | Ochre pieces | Use-wear traces on the pieces | SEM-EDXS<br>LA-ICP-MS | Non-invasive<br>Minimally invasive | None<br>None |
| Cavallo et al. [16] | Provenance (provenience) | Ochre pieces | None | Petrography<br>SEM-EDXS | Invasive<br>Invasive | Polished thin-sections<br>Polished thin-sections |
| Cavallo et al. [70,71] | Provenance (provenience), processing (heat treatment) | Ochre pieces | None | Powder XRD | Invasive | Grinding, powder |
| Cavallo et al. [32] | Heat treatment | Ochre pieces | None | Powder XRD<br>Petrography<br>SEM-EDXS<br>TEM | Invasive<br>Invasive<br>Invasive<br>Invasive | Grinding, powder<br>Polished thin-sections<br>Polished thin-sections<br>Grinding, suspension, one drop |
| d'Errico et al. [41]; Salomon et al. [72] | Evidencing human exploitation, heat treatment, provenance (geological origin) | Ochre lumps | Very small collection | SEM-EDXS<br>TEM<br>PIXE-PIGE<br>μXRD | Minimally Invasive<br>Minimally Invasive<br>Minimally Invasive<br>Minimally Invasive | Micro-samples, no preparation<br>Micro-samples, suspension, one drop<br>Micro-samples, enrobed in eopxy resin<br>Micro-samples |
| d'Errico et al. [73] | Raw material classification and selection | Ochre piece | Engraved piece | Visible spectroscopy<br>XRF | Non-invasive<br>Non-invasive | None<br>None |
| Dayet et al. [28,74] | Raw material classification and selection, heat treatment, provenance (geological origin), changes over time | Ochre pieces | Use-wear traces on the pieces | SEM-EDXS<br>XRD<br>Raman spectroscopy | Non-invasive<br>Non-invasive/invasive<br>Non-invasive | None<br>None/grinding<br>None |
| Dayet et al. [75] | Raw material classification and selection | Pigment pieces | Use-wear traces on the pieces | SEM-EDXS<br>μ-XRD<br>Raman spectroscopy<br>pXRF | Non-invasive<br>Non-invasive<br>Non-invasive<br>Non-invasive | None<br>None<br>None<br>None |
| Dayet et al. [17] | Provenance (provenience), group mobility | Ochre pieces/ferruginous materials | None | XRD<br>ICP-MS<br>ICP-OES | Invasive<br>Invasive<br>Invasive | Grinding, oriented powder<br>Grinding, dissolution<br>Grinding, dissolution |

**Table 1.** *Cont.*

| Reference | Archaeolgical Issue | Type of Remains | Sampling Limits | Method | Procedure | Sample Preparation |
|---|---|---|---|---|---|---|
| Dayet Bouillot et al. [27] | Raw material classification and selection, heat treatment, provenance (geological origin) | Ochre pieces | Use-wear traces on the pieces | SEM-EDXS<br>XRD<br>Visible spectroscopy | Non-invasive<br>Non-invasive<br>- | None<br>None<br>- |
| Dayet et al. [29] | Raw material classification, provenance (geological origin), changes over time | Pigment pieces | Museum collection | SEM-EDXS<br>XRD<br>pXRF | Non-invasive<br>Non-invasive<br>Non-invasive | None<br>None<br>None |
| Domingo et al. [61] | Identification of red pigments, processing and preparation | Pigment pieces and powders | None | SEM-EDXS<br>pXRF<br>XRD<br>FTIR<br>GC (organic part only) | ?<br>?<br>?<br>?<br>Likely invasive | ?<br>?<br>?<br>?<br>? |
| Eiselt et al. [59] | Provenance (provenience) | Ochre pieces and powders | None | NAA | Invasive | Grinding/settling, drying and grinding |
| Fiore et al. [34] | Technology of paint production | Ochre lumps and sediments | None | XRD<br>FTIR<br>SEM-EDXS<br>GC (organic part only) | ?<br>Invasive<br>?<br>- | ?<br>KBr disks<br>?<br>Crushing, dissolution |
| Garilli et al. [45] | Provenance (geological origin), heat treatment | Ochre sediments | None | SEM-EDXS<br><br>XRD<br><br>ATR-FTIR | Partialy invasive<br><br>Partialy invasive<br><br>Partialy invasive | Dried and "homogenized" (method not given)<br>Drying, "homogenized" (method not given)<br>Drying, "homogenized" (method not given) |
| Godfrey-Smith and Ilani [76] | Heat treatment | Hematite fragments | None | Thermoluminescence<br>XRD | Invasive<br>Invasive | Grinding, extraction of quartz grains<br>Grinding |
| Goemare et al. [47] | Provenance (geological origin) | Pieces of hematite | Use-wear traces on the pieces | HH-XRF<br>LA-ICP-MS<br>PIXE<br>XRPD | Non-invasive<br>Minimally invasive<br>Non-invasive<br>Invasive | None<br>None<br>None<br>Grinding, extraction of the clay fraction |
| Goemare et al. [77] | Provenance (geological origin) | Pieces of hematite | Use-wear traces on the pieces | HH-XRF<br>PIXE<br>XRPD<br>Petrography | Non-invasive<br>Non-invasive<br>Invasive<br>Invasive | None<br>None<br>Grinding, extraction of the clay fraction<br>Polished thin-sections |
| Henshilwood et al. [42] | Evidencing human exploitation, provenance | Lumps and micro-fragments of ochre | None | PIXE<br>SEM-EDXS<br>Petrography<br>μXRD | Invasive<br>Non-invasive/invasive<br>Invasive<br>Invasive | Polished thin-sections<br>None/polished thin sections<br>Polished thin-sections<br>Polished thin-sections |
| Hodgskiss, 2012 [26] | Raw material classification, properties and selection | Ochre pieces | Use-wear traces on the pieces | SEM-EDXS<br>FTIR<br>Raman spectroscopy | Non-invasive<br>Non-invasive<br>Non-invasive | None<br>None<br>None |

**Table 1.** *Cont.*

| Reference | Archaeolgical Issue | Type of Remains | Sampling Limits | Method | Procedure | Sample Preparation |
|---|---|---|---|---|---|---|
| Hovers et al., 2003 [24] | Provenance (geological origin) | Ochre lumps | None | Petrography<br>XRD<br>ICP-AES | Invasive<br>Invasive?<br>Invasive | Polished thin-sections<br>?<br>? |
| Hughes and Salomon, 2000 [78] | Classification, site comparison | Ochre pieces | None | Dosimetry<br>XRF<br>SEM-EDXS<br>TEM<br>XRD | Invasive<br>?<br>?<br>?<br>Invasive | ?<br>?<br>?<br>?<br>Grinding |
| Lebon et al., 2019 [52] | Raw material selection, links between different human activities | Ochre pieces, powders and residues | None | pXRF<br>SEM-EDXS<br>µXRD<br>Powder XRD | Non-invasive<br>Non-invasive<br>Non-invasive<br>Invasive | None<br>None<br>None<br>Grinding |
| MacDonald et al., 2011 (2013) [18,79] | Site comparison, provenance (geological origin) | Ochre samples | None | NAA | Invasive | Heat-sealed in high-purity polyethylene vials |
| MacDonald et al., 2018 [19] | Provenance (provenience) | Ochre pieces | Use-wear traces on the pieces | XRD<br>NAA | Non-invasive<br>None-invasive/invasive | None<br>None/grinding |
| MacDonald et al., 2020 [46] | Raw material properties | Sampling in an ochre mine | None | SEM-EDXS<br>XRD<br>NAA | Invasive<br>Invasive<br>Invasive | ?<br>Levigated<br>? |
| Matarrese et al., 2011 (Di Prado et al., 2007) [49,80] | Site comparison, provenance (geological origin) | Pigment pieces | None | Petrography<br><br>XRD | Invasive<br><br>Invasive | Polished thin-sections<br><br>Grinding, clay separation |
| Mathis et al., 2014 [36] | Provenance (geological origin), raw material selection | Ferruginous rocks | None | PIXE | Non-invasive? | None? |
| Mooney et al. [81] | Provenance (geological origin) | Archaeological ochre | None | Magnetic measurements | Non-invasive | None |
| Moyo et al. [44] | Raw material characterization, changes over time | Ochre pieces | Samples must not be damaged | XRF<br>XRD<br>FTIR<br>ICP-OES | Non-invasive/invasive<br>Invasive<br>Invasive<br>Invasive | None/grinding<br>Grinding<br>Grinding, KBr pellets<br>Grinding, digestion |
| Pierce et al. [64] | Provenance (provenience), changes over time | Hematite pieces | None | NAA | Invasive | Grinding |
| Pradeau et al. [37] | Provenance (provenience), changes over time | Pieces of coloring materials | None/pieces with use-wear traces not analyzed | Petrography<br>SEM-EDXS<br>XRD | Invasive<br>Invasive<br>Invasive | Polished thin-sections<br>Polished thin-sections<br>Grinding |
| Pomiès et al. [81] | Heat treatment | Ochre pieces | None | XRD<br>TEM | Invasive<br>Invasive | Grinding<br>Grinding, suspension |
| Popelka-Filcoff et al. [57] | Raw material classification | Ochre lumps and powders | None | NAA | Invasive | Grinding |
| Roebroeks et al. [4] | Evidencing human exploitation | Ochre sediments | None | XRD<br>SEM-EDXS | Invasive<br>Invasive | Grinding<br>Grinding |

**Table 1.** *Cont.*

| Reference | Archaeolgical Issue | Type of Remains | Sampling Limits | Method | Procedure | Sample Preparation |
|---|---|---|---|---|---|---|
| Salomon et al. [35] (Salomon [25]) | Raw material characterization, heat treatment | Pieces of coloring materials | None/pieces with use-wear traces not analyzed | Petrography<br>SEM-EDXS<br>XRD<br>FTIR<br>TEM | Invasive<br>Invasive<br>Invasive<br>Invasive<br>Invasive | Polished thin-sections<br>Grinding<br>Grinding<br>Grinding<br>Grinding, suspension |
| Salomon et al. [82] | Heat treatment | Pieces of ferruginous rocks | None | μ-XRD<br><br>SEM-FEG<br><br>TEM-FEG | Non-invasive/minimally invasive<br>Non-invasive/minimally invasive<br>Minimally Invasive | None/micro-samples<br><br>None/micro-samples<br><br>Micro-samples? |
| San Juan-Foucher [83]; Pomiès and Vignaud [84] | Raw materail selection, heat treatment | Pieces of coloring materials | None | XRD<br>TEM | Invasive<br>Invasive | Grinding<br>Grinding, suspension |
| Scadding et al. [62] | Provenance (provenience), raw material selection | Ochre manuports | None | LA-ICP-MS | Minimally invasive | None |
| Tortosa et al. [48] | Provenance (provenience), changes over time | Ochre fragments | None | XRD<br>ICP-MS<br>Petrography<br>SEM-EDXS<br>XRF | Invasive<br>Invasive<br>Invasive<br>Invasive<br>Invasive | Grinding<br>Grinding<br>Polished thin sections<br>Polished thin sections<br>Grinding |
| Trabska et al. [65] | Provenance (provenience) | Ferruginous artifacts | None | PIXE<br>XRF | ?<br>? | ?<br>? |
| Velliky et al. [85] (Velliky et al. [86]) | Provenance (geological origin), changes over time | Pigment pieces | None | NAA<br>XRD<br>SEM-EDXS | Invasive<br>Invasive<br>Non-invasive? | ?<br>Sub-sampling, grinding<br>None? |
| Wadley [87] | Taphonomic analysis, accidental heating | Ochre powders | None | XRD<br>XRF<br>ICP-MS<br>Micro-morphology | Invasive<br>Invasive<br>Invasive<br>Invasive | Grinding<br>Grinding<br>Grinding<br>Polished thin sections |
| Zarzycka et al. [63] | Provenance (provenience), group mobility | Ochre nodules | None | ICP-OES | Invasive | Grinding, dissolution |
| Zilhao et al. [50] | Evidencing human exploitation | Pigment lumps and residues | None | Powder XRD<br>SEM-EDXS<br>Raman spectroscopy | Invasive<br>Minimally invasive<br>Minimally invasive | Grinding?<br>Micro-sampling<br>Micro-sampling |

Choosing between the invasive and non-invasive mode for ochre analysis is more delicate than just choosing a method among a list. As a reminder, by "invasive" we mean that a bit of matter is removed from the object or the sample analyzed, while by "non-invasive" we mean that no matter is removed or destroyed during analysis (totally non-destructive). There are methods that are designed for non-invasive analyses, but they are also often designed for homogenous flat samples, not for heterogeneous and irregular geomaterials. Several methods can be employed in both manners, but the non-invasive mode is often less accurate and/or less sensitive. Each non-invasive technique has its disadvantages in comparison with the equivalent invasive technique. As a consequence, certain archeological issues are still widely addressed by using invasive methods (see Table 1). The main advantages and disadvantages of each non-invasive technique are reviewed here for readers that would not be familiar with their utilization. The methodology developed for the most common archeological issues in ochre studies are described in the following sections.

### 2.2. Non-Invasive Methods Used in Ochre Studies: A Rreview of Their Limits

Micro-Raman spectroscopy is the most common analytical technique designed for non-invasive analyses. Samples of all sorts and forms, from whole objects to powders, can be placed under a microscope coupled with a Raman spectrometer. The laser beam used in Raman analysis is usually non-destructive for geomaterials. However, oxy-hydroxides such as goethite are very sensitive to the heat induced by the laser, and at high beam power they are turned into hematite [88]. As a consequence, very low laser power is recommended for the analysis of iron oxy-hydroxides in general. Moreover, a spectrum is representative of a very small volume (particle with a diameter of <1 μm to 5 μm with a ×100 objective, more than 5 μm for ×50 objective; [89]), meaning that multiple measurements must be made for large pieces in order to guarantee a representative sampling. Such a necessity may be time consuming for the analysis of a large corpus of pieces, and furthermore makes sample comparison sometimes difficult.

SEM-EDXS, PIXE, micro-XRF, or portable XRF are widely used in non-invasive elemental analyses. The geometry of the samples, however, influences the accuracy of the semi-quantitative (EDXS with classic standards, XRF following the fundamental parameter method) or quantitative (PIXE with dedicated standards; XRF using empirical calibration curves) results. They are more accurate when performed on flat surfaces [90–95]. SEM-EDXS analyses are superficial (the penetration depth is dependent on the atomic number, usually <10 μm) and semi-quantitative. The scanning mode allows for the analysis of areas of several hundred micrometers, but also of small particles of no more than 1 or 2 μm [96]. This makes SEM-EDXS analyses ideal for the characterization of a matrix or cement that is composed of clay particles with silt and sand-sized inclusions dispersed in it (combination of both semi-quantitative and qualitative approaches; [16,27,37,42,46,74]). Proton beams have a higher penetration depth, but it is still lower than the penetration depth of an X-ray beam [95,97]. PIXE analyses are more sensitive and allow for the quantification of several trace elements, from the transition metals to some rare earth elements. SEM-EDXS and PIXE are designed for the analysis of small surfaces at micrometric scales (not more than 1 mm$^2$; [98–100]). They can be used for elementary analyses of thin layers because of the low critical penetration depth of secondary X-ray photons produced by primary electron or proton beams [95–97].

The penetration depth of X-ray beams in XRF instruments is dependent on the atomic number of the analyzed material, the anode of the X-ray tube, and the voltage applied [101]. The spatial resolution of XRF analyses, except when using micro-XRF methods, is higher than that of SEM-EDX and PIXE analyses, as the XRF beam is more than 1 mm in diameter and is usually of several millimetres in diameter for portable instruments [98–100]. This higher beam size is advantageous for global analyses of large areas. Nonetheless, a high number of measurements is recommended for the analysis of heterogeneous rocks in order to be assured a high precision [101]. Moreover, matrix effects can be very strong in XRF

analyses, making the quantification of elements complex and time consuming when the samples analyzed have very different mineralogical matrixes (see e.g., [102,103]). The set of trace elements that can be quantified is very variable. XRF can also be carried out at a microscopic scale (micro-XRF), allowing for analyses with a higher spatial resolution, yet the penetration depth of the X-ray beam still remains higher than for other bulk analyses.

In XRD analyses, randomly oriented and oriented powders are the most common preparation modes for heterogeneous inorganic materials. The random mode ensures that the crystals will diffract in random directions and that the diffracted signal collected by the rotating detector will be representative of the proportions of each phase in the sample (no higher intensities due to crystal orientation; [104,105]). Oriented powders are applied on glass plates in order to orient platy particles such as clay minerals. The intensity of clay minerals' main peaks increases, which improves the detection sensitivity and the identification precision of phases from this large mineralogical family [106,107]. Cohesive samples can also be analyzed by classic XRD using Bragg–Brentano geometry. In cases where the crystals are not randomly oriented within the geomaterial, however, XRD peaks whose position do not match that of the rotation circle of the detector will not be detected. The signal collected may therefore not be representative of the overall sample composition. Moreover, if the sample is not flat, angular offsets can be observed between the diffraction signal of the lower and the higher reliefs. This effect can be eliminated by using a parallel beam geometry [74,104]. In micro-XRD, the use of two-dimensional area detectors allows for in situ analyses of minerals within cohesive rocks and polished thin sections, whatever the orientation of the crystals [104]. Micro-XRD analyses are also used for the study of micro-samples in ochre studies [35,72,82].

FTIR analyses were not originally designed for non-invasive characterization. The transmission mode requires the analyst to make KBr pellets in which powder samples are dispersed [108]. A more recent FTIR technique, the attenuated total reflectance (ATR) mode, allows for the analysis of very small amounts of powder or micro-samples. The latter will however be crushed during analysis by the micrometer-controlled compression clamp that is used to make contact between the sample and the crystal, which is a prerequisite to creating an attenuated total reflectance signal [109]. In both cases, the signal collected consists of the vibrational bands of the molecules composing the sample. Non-invasive analyses can only be made in the reflection mode, but the signal obtained is different and more complex. When the reflection mode is used, the vibrational bands of the minerals are not systematically detected. Other parameters than the molecular composition influence the signal in ways that are not predictable (infrared distorsion; [110,111]). Only a dedicated referential database, with examples analysed in the same conditions, can allow an accurate interpretation of the spectra and the identification of the minerals present in an unknown sample.

Some invasive methods have been enhanced in order to allow minimally invasive measurements. This is the case with ICP-MS, which can be coupled with a laser ablation system (LA-ICP-MS). The ablation allows for the analysis of very small volume samples, as it forms a small hole with a diameter of some hundreds of micro-meters at the surface of a cohesive sample [1,47,62]. The high spatial resolution of this method can be an advantage or a disadvantage depending on whether a "grain per grain" or global analyses is preferred. In order to counterbalance ochre materials' heterogeneity, some studies propose to carry out multiple measurements per sample [62].

### 2.2.1. Raw Material Characterization of Ochre: Macroscopic Examination versus Physico-Chemical Analyses

Several ochre studies dealing with ochre characterization do not include physico-chemical analyses [21,23,30,33,40,112], or include few physico-chemical analyses with few inferences on raw material characterization [26,113]. Using macroscopic and microscopic examination alone is usually avoided in mineralogical and petrologic studies for several reasons, that could appear trivial when dealing with advanced geological issues: this approach is qualitative; some of the criteria might be subjective; the surface of the rock

might not be representative of the inside (presence of a patina, a layer of alteration); the identification of coarse grains or large crystals within a rock is not always possible; and the fine-grained fraction cannot be properly characterized. Why in this case physico-chemical analyses are not systematically conducted in ochre studies dealing with ochre raw material characterization? There might be a general reason for this choice too, that might appear trivial for archeologists on the other side. Physico-chemical analyses require the access to dedicated facilities and are time consuming. When raw material characterization is not the main goal of the research or might not require a high precision as regard to the research question addressed, macroscopic examination revealed to bring relevant results.

For instance, Watts [23] used several qualitative but strictly defined macroscopic features in order to look for selection criteria amongst raw materials and to question what was the purpose of ochre use for Middle Stone Age people from Pinnacle Point (South Africa). Because he gave very precise definition of each criterion and compares their distribution throughout the sequence independently, his study is reproducible (in theory) and the reliability of the results can be evaluated. Another successful macroscopic study is found in Mauran et al. [33], who inventoried and described various ochre remains (ochre pieces, red deposits on beads, ochre grinding tools, etc.) from the Later Stone Age (LSA) site of Leopard cave (Namibia). Their large but strictly defined categories of ochre raw materials for the characterization of ochre pieces ("iron oxide nodules" versus "various ferruginous rocks") are just one element among others allowing to discuss the different utilizations of ochre at the site. However, when each raw material category is too accurate to be properly identified from macroscopic criteria (based on silt versus clay texture for instance) and if they are not strictly distinguished one from each other by macroscopic criteria, there is a real risk of inaccuracy. The only way to increase the robustness of the classification is to carry out more accurate mineralogical, petrographic, and elemental analyses. On the other hand, physico-chemical characterization presents a major inconvenience: it requires making a selection of samples. Indeed, the number of pieces within an archeological assemblage is usually high (dozens, hundreds of pieces, sometimes thousands) and certain pieces cannot be destroyed (cultural heritage value). This means that the most accurate analytical methods used in mineralogy and geochemistry cannot be applied to a complete assemblage. Macroscopic examination remains one of the most direct way to characterize a large corpus of archeological ochre pieces.

A final point about ochre raw material characterization concerns the ochre materials themselves. Ferruginous geomaterials present specific features making their characterization complex and difficult in comparison with other families of geomaterials. Iron oxide and oxy-hydroxdes commonly form under weathering conditions, which means that there might not be a clear separation of composition between the host rocks and the iron-enriched pedomaterials that are associated to them (see e.g., [15,114,115]). Considering that iron oxides and oxy-hydroxides can also formed within sedimentary rocks, under hydrothermal alteration, in igneous rocks, and can also be found in metamorphic rocks, ferruginous geomaterials are highly diverse and might present heterogeneous petrological features. Their petrological characterization is not trivial. As a consequence, making "raw materials" categories from a collection of ochre pieces is always tricky and there are at least two points that shall be considered in this process according to current research:

- The choice of the geological terms used to name the categories, the best terms probably being those used by geologists in the region of interest (refer to literature on the regional geology; see e.g., [21,26]).
- The fact that intermediate forms between hosts rocks and associated weathering products exist and make absolute classification difficult (see e.g., [17,28]).

2.2.2. Ochre Provenance Research

Provenance research allow us to interrogate both the acquisition and circulation of archeological artifacts in various ways. In ochre studies, such approaches tend to be oriented towards: (1) the seriation of archeological pieces in order to discuss their geological

origin; (2) the finding of direct petrological and mineralogical proxies for determining geological origin; (3) the discrimination between geological sources to facilitate future archeological investigations; (4) the comparison of archeological samples with a geological reference collection (provenance or provenience studies *sensu stricto*). The difference between provenance and provenience is discussed elsewhere [39]. We will favour the term "provenience" as proposed by Zipkin et al. [38,39] when referring to issue 4, although the term is not widespread in ochre studies yet. The expression "geological origin" is preferred when sampling allows for only an approximate localization of sources.

Two main approaches have been proposed to investigate ochre provenience and geological origins. In certain cases, petrographic and mineralogical analyses are dominant, while in others, and more frequently moreover, geochemical analyses were favored (Table 2). In the current state of research, these methods are almost all invasive. For example, an in-depth petrological and mineralogical provenance study was carried out by Pradeau et al. [37] on red coloring materials from two Neolithic sites in southeastern France. They used SEM-EDXS analyses on thin sections along with powder XRD analyses in order to identify differences in the rock fabric and the mineralogical composition within a wide range of geomaterials from the region. They were able to distinguish between both local and extra-local sources and used this criterion to discuss the economic patterns observed at the two sites.

**Table 2.** Short description of the various methodologies developed in ochre provenance researches in the last two decades.

| Reference | Context | Methods | Mode | Main Goal | Arch. Samples | Geol. Samples | Data Treatment |
|---|---|---|---|---|---|---|---|
| Barham [2] (Young [66]) | Twin Rivers, Zambia | XRF, ICP-MS, SEM-EDXS | Invasive | Seriation, geol. origin | 7 | - | Qualitative |
| Hovers et al. [24] | Qafzey, Israël | Pétrography, XRD, ICP-AES | Invasive | Seriation, geol. origin | 71 | 7 | Qualitative |
| Mooney et al. [81] | Australia | Magnetic parameters | Non-invasive | Provenience | 2 | 8 | Qualitative |
| Kiehn et al. [116] | Botswana | NAA | Invasive | Source discrimination | - | 72 | Multivariate statistical analyses |
| Popelka-Filcoff et al. [56] | Missouri, USA | NAA | Invasive | Source discrimination | - | 69 | Multivariate statistical analyses |
| Popelka-Filcoff et al. [57] | Jiskairumoko, Perou | NAA | Invasive | Seriation | 65 | - | Multivariate statistical analyses |
| Trabska et al. [65] | Dzierżysław 35, Poland | PIXE, TXRF | ? | Geol. origin | 19 | 11 | Multivariate statistical analyses |
| Bernatchez [69] | Nelson Bay Cave, South Africa | XRD, PIXE | Invasive | Seriation | 54 | - | Qualitative |
| Popelka-Filcoff et al. [117] | Arizona, USA | NAA | Invasive | Source discrimination | - | 110 | Multivariate statistical analyses |
| Iriarte et al. [118] | Tito Bustillo and Monte Castillo, Spain | Petrography, XRD, SEM-EDS, ICP-MS | Invasive | Source discrimination | - | 24 et 24 | Qualitative |
| Salomon [25]; Salomon et al. [35] | Arcy-sur-Cure, France | Macroscopic examination, SEM-EDS, XRD, petrography | Minimally invasive | Seriation, geol. origin | 100 | - | Qualitative |
| d'Errico et al. [41]; Salomon et al. [72] | Es Skhul, Israël | XRD, SEM-EDS, PIXE | Minimally invasive | Geol. origin | 4 | - | Qualitative |
| Eiselt et al. [59] | Arizona, USA | NAA | Invasive | Provenience | 25 | 54 | Multivariate statistical analyses |
| MacDonald et al. [18] | Canada | NAA | Invasive | Seriation, geol. origin | 3 | 61 | Multivariate statistical analyses |
| Attard Montalto et al. [60] | Malta | Petrography, ICP-AES | Invasive | Provenience | 21 | 58 | Multivariate statistical analyses |
| Beck et al. [67] | Arcy-sur-Cure, France | PIXE | Non-invasive | Seriation, geol. origin | 27 | - | Qualitative |
| Popelka-Filcoff et al. [58] | Austalia | k0-NAA | Invasive | Source discrimination, comparison between methods | - | 100 | Multivariate statistical analyses |

**Table 2.** *Cont.*

| Reference | Context | Methods | Mode | Main Goal | Arch. Samples | Geol. Samples | Data Treatment |
|---|---|---|---|---|---|---|---|
| Kingery-Schwartz et al. [119] | North America, USA | XRD, pXRF, NAA | Non-invasive, invasive | Source discrimination | - | ? | Qualitative, multivariate statistical analyses |
| Mathis et al. [36] | Ormesson, France | PIXE | Non-invasive | Geol. origin | ? | 29 | Bivariate analysis |
| Zipkin et al. [38] | Northern Malawi | NAA, LA-ICP-MS | Invasive | Source discrimination, comparison between methods | - | 22 | Multivariate statistical analyses |
| Dayet et al. [17] | Diepkloof rock shelter, South Africa | XRD, ICP-OES, ICP-MS | Invasive | Geol. origin, provenience | 28 | 80 | Qualitative, multivariate statistical analyses |
| Pradeau et al. [37] | Pendimoun and Giribaldi, France | SEM-EDS, XRD | Invasive | Geol. origin | 56 | ? | Qualitative |
| Cavallo et al. [16] | Fumane cave and Tagliente Rockshelter Italia | Petrography, SEM-EDXS | Invasive | Geol. origin | ? | 66 | Qualitative |
| Cavallo et al. [70] | Fumane cave and Tagliente Rockshelter Italia | Powder XRD | Invasive | Geol. origin | ? | - | Semi-quantitative |
| Dimuccio et al. [120] | Grotta della Monica, Italia | pXRF, XRD, Raman, FTIR | Invasive | Source characterization | - | 81 | Qualitative, multivariate statistical analyses |
| Zipkin et al. [121] | Central Kenya | LA-ICP-MS | Invasive | Source discrimination | - | 43 | Multivariate statistical analyses |
| MacDonald et al. [19] | Haney Cook and Ball villages, Northern America | XRD, NAA | Invasive | Provenience | 23 | - | Qualitative, multivariate statistical analyses |
| Velliky et al. [86] | Southwestern Germany | NAA | Invasive | Source discrimination | - | 139 | Multivariate statistical analyses |
| Zarzycka et al. [63] | La Prele Mamoth, USA | ICP-OES | Invasive | Provenience | 7 | 24 | Multivariate statistical analyses |
| Peirce et al. [64] | Central Missouri, 4 archaelogical sites, USA | NAA | Invasive | Provenience | 38 | 69 | Multivariate statistical analyses |
| Velliky et al. [85] | Hohle Fels, Geißenklösterle and Vogelherd, Germany | NAA, XRD, SEM-EDS | Invasive | Provenience, geol. origin | 210 | 139 | Qualitative, multivariate statistical analyses |
| Zipkin et al. [39] | Central Kenya | LA-ICP-MS | Invasive | Source discrimination | - | 36 | Multivariate statistical analyses |
| Mauran et al. [122] | Leopard cave, Namibia | ICP-OES, ICP-MS | Invasive | Geol. origin | 41 | 94 | Multivariate statistical analyses |

Legend—Geol.: geological; Arch.: archaeological.

The main methods used for investigating major and trace element composition in recent ochre provenance studies were summarized in [39]. Mathis et al. [36] proposed the first complete non-invasive ochre provenance study using a consistent strategy of geological sampling based on strictly defined petrological and contextual criteria (type of rock, geological formation) and PIXE analyses on cohesive unprepared archeological samples. They showed that the iron-rich concretions from the Middle Paleolithic site of Ormesson (France) were collected in a geological formation whose outcrops can be found at no more than 5 km from the site. However, this could represent a particular case given that the number of elements used in the discrimination is low in comparison with the large set of elements used in other provenance studies.

Amongst invasive geochemical approaches, the work from MacDonald et al. [19] is a thorough illustration of how a comprehensive ochre provenance study can be designed. They studied ochre pieces from two Paleoindian sites from the Great Lakes (North America) and compared them to a wide geological collection constituted of samples collected from 10 iron-oxide bearing deposits chosen according to ethnographic, historical, and archeological criteria. They began with petrologic observation and XRD analyses in order to better

characterize the sources. They carried out NAA analyses in order to obtain a large set of trace elements, and they then used multivariate statistical analyses for the discrimination of the elemental signature of the sources. Using the three first principal components of a principal component analysis (PCA) projected on biplots they show that the sources can be graphically discriminated. The projection of archeological samples within these models allow for the identification of their provenience, or at least a geological origin, for the majority of them.

Data treatment is crucial when dealing with the statistical analysis of elemental quantitative results. For ochre studies, there has been some debate on whether elemental data should be log-transformed or not before their use in multivariate analyses [19,38,79,121,122]. It is acknowledged that the use of log-ratios is more suitable when dealing with compositional data [123,124]. Several authors employed log-ratios to iron, following the pioneering work of Popelka-Filcoff and colleagues [56,117]. The main advantages of this data transformation are discussed elsewhere [17,19,39,56,64]. Their interest may however vary according to the choice of the elements that are included in the statistical analysis. There is no consensus on how to select elements in order to allow for the most neutral and efficient discrimination. Looking for the highest degree of discrimination between sources by removing elements following purely statistical procedures when only a limited geological sample is studied might lead to an underestimation of intra-source variability. In such an instance, an archeological sample might not match with a source, despite its geochemical association with it. Putting all elements in the analysis without looking for their relevance in inter-source discrimination might appear to be the most neutral way to avoid such a bias, yet such a practice leads to other biases. In such conditions, the signature of the sources might be very broad and based on very common major or trace elements. An archeological sample might match with such a reference just because it comes from the same geological formation or because it is composed of the same assemblage of minerals.

The method proposed from the beginning by Popelka-Filcoff et al. [56] appears to be a good compromise, as they propose that the most reliable signature of a source is the signature of the iron oxide or oxy-hydroxyde phases. They select elements that are positively correlated with iron. Nonetheless, some studies where this method was tested found that no elements were positively correlated with iron when all the sources were included in the process [17,19]. Another limit of this method, which may also explain the first limit discussed, is that iron oxides with high Fe-content are not directly comparable with iron-bearing rocks, as the latter have significantly lower iron contents [17,64]. The use of log-transformed Fe-ratios does not provide a sufficient work-around that smooths the large variations in iron between them. This might be due to geochemical trends that are not entirely understood yet, such as non-linear element migrations, or recrystallization above a threshold of iron concentration, etc. Context-dependent solutions were developed when these limits were encountered [17,19,64].

### 2.2.3. Iron Oxy-Hydroxide Heat Treatment

While ethnographic data suggest that a heat treatment may be part of the operatory sequence of technical gestures preceding ochre use [125], several archeological studies proved this hypothesis to be valuable since Paleolithic times [32,82,126]. The main goal of heat treatment is a change of color. When yellow goethite is heated at a temperature ranging between 260 °C and 300 °C, it turns into red hematite by dehydration:

$$2\ \alpha\text{FeOOH} \rightarrow \alpha\text{Fe}_2\text{O}_3 + \text{H}_2\text{O}. \tag{1}$$

This is how yellow ochre is converted into red ochre. This well-described reaction is not an oxidation of the goethite phase—the degree of oxidation remains the same—and it is topotactic at a first stage, which means that it does not involves major structural modifications [126–131]. The reaction starts at about 200 °C and is complete at 300 °C. During the process, iron atoms migrate within the octahedral network (position of the oxygen atoms) which results in a "disorder" in the overall structure of the newly formed

hematite. Simultaneously, nano-pores develop inside the crystals where water vapor escapes. When the temperature increases, the structure levels off and pores disappear [43,126]. A recrystallization occurs at temperatures higher than 700 °C.

The main proxies of this transformation are: (1) an anisotropic broadening of hematite X-ray diffraction peaks when the heat temperature is lower than 600 °C; (2) the presence of nano-pores within the precursor goethite crystals observable under SEM-FEG (Field Emission Gun) and TEM (Transmission electron microscopy) techniques [32,43,82,126,130,131] (Table 3). The structural "disorder" of heated goethite is also detected in Raman spectrometry [43,132]. However, the band relating to heating conditions—at about 650–660 cm$^{-1}$—is observed in several natural hematites and was interpreted in different ways depending on the authors [132,133]. An anisotropic broadening of X-ray peaks was also detected in some natural hematite samples [126,129]. Whether these samples could have been submitted to incidental heating as in a wildfire for instance has not been investigated yet. Another reaction occurs in goethite if heated in the presence of organic matter. In these very specific conditions maghemite, an iron oxide of an orange-red color ($\gamma Fe_2O_3$), is formed [126]. The presence of this iron oxide was also used as a proxy of heating treatment [82,126].

In-depth studies of these different proxies in archeological ochre pieces can be found in, e.g., Gialanella et al. [43], Cavallo et al. [32] and Salomon et al. [82] (see Table 3). The authors of the latter carried out non-invasive and minimally invasive micro-XRD and SEM-FEG analyses along with TEM observations on small micro-samples from several French Upper Paleolithic sites. In particular, they demonstrated that goethite was intentionally heated to produce hematite at the Solutrean site of Les Maîtreaux (France) by showing that the inside and outside of large blocks of hematite show similar patterns of X-ray peak broadening. They estimated the cycle of temperature required to obtain such proxies to be up to 300 to 400 °C for at least two hours. The location of these pieces far from any known combustion feature at the site, in close association with a pile of flint, was further evidence of a heat treatment. Nonetheless, differentiating between intentional and incidental heating is more often a matter of archeological context than a physico-chemical issue (see also [25]).

**Table 3.** Short description of the various methodologies developed for the study of heat treatment features in ochre and iron-bearing geomaterials.

| Reference | Context | Arch. Samples | Methods | Experimental Samples | Conditions of Heating Experiments |
|---|---|---|---|---|---|
| Onoratini and Perinet [134] | 13 Paleolithic sites from south-east France | 60 | XRPD | 11 nat. goethite | |
| Pomiès et al. [126,129] | 5 Paleolithic sites from France | 30 + 15 | XRPD, TEM | Syn. goethite, 1 nat. goethite | Oven in air 250 to 1000 °C |
| Baffier et al. [135] | Arcy-sur-Cure, France | 3 | XRPD, TEM | - | - |
| Pomiès et al. [136] | Lacaux, France | 4 | XRPD, TEM | - | - |
| Godfrey-Smith et Ilani [76] | Qafzeh Cave, Israël | 4 | TL | - | - |
| Pomiès and Vignaud [84]; San Juan-Foucher [83] | Bois-Ragot, France | 14 | XRPD, TEM | - | - |
| Lahaye [137] | La Honteyre, France | 4 | TL | - | - |
| Salomon [25] | Arcy-sur-Cure, France | 70 | µXRD, TEM | ? | ? |
| Salomon [25]; Salomon et al. [82] | Combe-Saunière 1, France | 13 | µXRD, SEM-FEG, TEM | ? | ? |
| Salomon [25]; Salomon et al. [82] | Les Maîtreaux, France | 24 | µXRD, SEM-FEG, TEM | ? | ? |
| Salomon [25]; d'Errico et al. [41]; Salomon et al. [72] | Es-Skhul, Israël | 4 | µ-XRD, TEM | ? | ? |
| Gialanella et al. [43] | Riparo Delmari, Italia | 6 | XRPD, TEM, Raman | 3 nat. goethite | Furnace in air 1000 °C |
| Salomon et al. [82] | Grotte Blanchard | 6 | µXRD, SEM-FEG, TEM | ? | ? |
| Dayet et al. [27] | Klasies river main site | 39 | XRD on surfaces | - | - |
| Cavallo et al. [32] | Fumane cave, Italia | - | XRPD, TEM | - | - |
| Cavallo et al. [32] | Tagliente rockshelter, Italia | - | XRPD, TEM | - | - |

Legend—Arch.: archaeological; Nat.: natural; Syn.: synthetic.

Another method was used more seldomly in case studies for investigating heating features in iron-bearing rocks. Thermoluminescence (TL) techniques are widely known for their capacity to detect the last heating event of a material and to evaluate the age of this event [138]. The same techniques can be applied to determine if an ochre piece was heated if it contains thermoluminescent crystals such as quartz [76,137]. However, again, clear evidence of an intentional heat treatment may not be found.

## 3. Non-Invasive Analyses: A Case Study from Diepkloof Rock Shelter, South Africa

Non-invasive methods, as we showed, present different sorts of limits that have to be considered when choosing them but also when interpreting the results which they provide. This is especially true for archeological artifacts because of a second order of possible biases in non-invasive analyses, that is the preservation of their composition. Their surface in particular might have been submitted to subtle but consistent changes in mineralogical and elemental composition due to various post-depositional phenomenon. In order to address these two methodological aspects, dedicated research can be undertaken.

Here we present a methodological study that was carried out in the framework of a wider research project on the characterization of ochre remains of Diepkloof rock shelter, Western Cape. The main goals of this project were to determine: (1) what kind of raw materials were selected and used; (2) how they were processed; (3) from where and how raw materials were collected; (4) how modalities of ochre exploitation and use vary throughout the sequence. The main constraint we faced was the difficulty of conducting invasive analyses because of the high proportion of ochre pieces showing evidence of use and because of the cultural heritage value of the Middle Stone Age (MSA) collections of Diepkloof rock shelter as a whole. The first steps of the project consisted in performing entirely non-invasive analyses of a large selection of archeological ochre pieces [28,74]. In parallel, when a consistent geological collection of reference was constituted, tests were conducted in order to evaluate the accuracy, the sensitivity, and relevance of the non-invasive protocols used to characterize the archeological samples by comparing them to more conventional bulk and structural invasive analyses. A final step consisted of the selection of a small but representative corpus of 28 archeological ochre pieces for invasive analyses in order to evaluate the influence of post-depositional alterations on the composition of their surfaces, and to evaluate the relevance of previously obtained non-invasive results. These methodological steps are partly described in a PhD thesis [139].

Three methods that can be used in non-invasive and invasive modes were chosen:

- XRD: for structural analyses;
- SEM-EDXS: for bulk analyses of major elements; and
- PIXE: for bulk analyses of major and traces elements.

XRD and SEM-EDS methodological investigations were taken into account in all previously published articles, although they were never properly described [17,27,28]. The PIXE results were never published before.

### 3.1. Material and Methods

#### 3.1.1. Archeological Samples

Diepkloof rock shelter lies on the west coast of South Africa, about 200 km north of Cape Town. This large quartzitic sandstone shelter overlooking the small Verlorenvlei River valley about 14 km from the present shoreline has been investigated since 1999 by a South African-French team led by J.-P. Rigaud, P.-J. Texier, and C. Poggenpoel. Excavated in three different sectors, the sequence reaches a depth of 3.1 m in the 3 m$^2$ "main sector" (squares K6, M6, N6) and is mainly composed of MSA deposits [140,141]. The lower stratigraphic units are associated with an estimated date of 110 ky, while the overlying MSA units have been dated to 55 ky by a series of Optically Stimulated Luminescence (OSL) and Thermoluminescence dating [142]. The whole collection of pieces from the main sector (squares M6 and N6) were analyzed in detail [28].

The 27 ochre pieces selected for invasive analyses were either chosen because they had a typical composition among the overall corpus (typical shale, typical ferricrete) or because their composition was intriguing and hard to interpret. Although some pieces with use-wear traces enter in the second category, no piece with clear evidence of use have been destroyed. No other archeological criterion was used to make the selection, although further questions emerged during the provenance research undertaken. Sample selection is always a compromise. Further description of these pieces can be found in [17,28].

### 3.1.2. Geological Samples

The geological samples used in this study come from 12 different sources of raw materials that were collected during several campaigns of survey in a diameter of more than 50 km around Diepkloof rock shelter (Table 4). Five of these sources are shale outcrops (Shale 1 to Shale 5), two are phyllite outcrops that were associated to shale sources (Shale 6 and Shale 7), four are ferricrete from sub-primary deposits (Ferr 1 to Ferr 4), and the last one is associated to a ferricrete deposit (Ferr 2) but it is constituted by indurated shale. Most of these deposits are described elsewhere [17,28]. For the methodological research presented here, 24 samples chosen among all the collected sources were analyzed by XRD (surface, oriented powders) and 5 of them (3 shale/phyllite fragments and 2 ferricrete nodules) were analyzed by EDXS and PIXE (surface, powder pellets).

**Table 4.** Geological outcrops sampled for ochre raw material characterization and provenance studies at Diepkloof rock shelter. Samples from this geological collection were used to investigate the limits of non-invasive application of SEM-EDXS, XRD, and particle-induced X-ray emission (PIXE) analyses on cohesive pieces. Legend—Geol.: geological; Ref.: reference.

| Source | Rock Type | Geol. Formation | GPS Coordinates | Nearest Town | Ref. |
|---|---|---|---|---|---|
| Shale 1 | Shale | Table Mountain | 32°23′13″ S 18°27′10″ E | Elands Bay | 14040 to 14055 |
| Shale 2 | Shale | Klipheuwel | 32°18′57″ S 18°21′18″ E | Elands Bay | 14691 |
| Shale 3 | Shale | Klipheuwel | 32°27′30″ S 18°30′55″ E | Redelinghuys | 14693 |
| Shale 4 | Shale | Klipheuwel | 32°29′32″ S 18°33′47″ E | Redelinghuys | 14694 |
| Shale 5 | Shale | Table Mountain | 32°34′23″ S 18°43′50″ E | Het Kruis | 14692 |
| Shale 6 | Phyllite | Malmesbury | 32°31′21″ S 18°37′56″ E | Redelinghuys | 14696 |
| Shale 7 | Phyllite | Malmesbury | 32°52′36″ S 18°46′02″ E | Piketberg | 14700 |
| Ferr 1 | Ferricrete | Tertiary/quaternary | 32°31′21″ S 18°37′56″ E | Redelinghuys | 14697 |
| Ferr 2 | Indurated shale | Tertiary/quaternary | 32°42′08″ S 18°49′46″ E | Eendekuil | 14698 |
| | Ferricrete | Tertiary/quaternary | 32°42′08″ S 18°49′46″ E | Eendekuil | 14699 |
| Ferr 3 | Ferricrete | Tertiary/quaternary | 32°52′25″ S 18°47′14″ E | Piketberg | 14701 |
| Ferr 4 | Ferricrete | Tertiary/quaternary | 32°07′16″ S 18°26′30″ E | Lamberts Bay | 14337 |

### 3.1.3. Preparation of the Samples

Archeological and geological cohesive pieces were carefully washed by using distilled water and a toothbrush in order to remove all forms of external deposits on them (sediments, soluble salts). For archeological samples, some soluble salts were still detected on the surface of part of the samples despite intense washing. In order to remove the deposits and their superficial part (patina, cortex, any superficial layer which composition might be different from the inside of the piece), the surface of the pieces was abraded against a diamond saw. They were then crushed into powder into an agate mortar. A small quantity of each powder sample was mixed with water and then placed on a glass slide and dried in a 50 °C oven (oriented powder). For EDXS and PIXE analyses, pellets were made. Geological samples were submitted to the exact same protocol.

Before they were ground, a fragment of 7 archeological pieces was extracted by cutting it with a diamond saw. It was enrobed in an acrylic resin under vacuum in order to make polished sections.

### 3.1.4. XRD Analyses

Structural phases were determined by X-ray diffraction. Data were collected with a Bruker D8 Advance diffractometer, equipped with a PSD Linxeye detector and operating with a Cu Kα radiation (λ = 1.5405 Å). Parallel beam geometry (obtained with a Göbel mirror) was used to carry out surface analyses. Bragg–Brentano geometry was used to

analyze oriented powder samples. A 0.3 mm divergent slide was positioned after the X-ray tube with a "knife" placed above the samples in order to cut incident X-rays at low angles and create optimal conditions for detecting clay minerals. The identification of the phases and the semi-quantitative evaluation were done with EVA (Bruker) software (Diffrac.suite). Concentrations calculated in this manner are not truly representative of the proportions of the phases but they are the best proxy we can use to compare the samples. For surface analyses, only the presence/absence of the phases were estimated.

### 3.1.5. SEM-EDXS Analyses

A JEOL 6460 LV SEM (Oxford instruments) instrument was used, equipped with a low vacuum system which allows the imagery and analysis without specific preparation (coating) of the sample. For cohesive samples and powder pellets, we used the *low vacuum* mode with a pressure of 20 Pa and an acceleration voltage of 20 kV. For polished thin sections, a carbon coating was deposited on their surface and EDXS analyses were carried out in *high vacuum*. Semi-quantitative analyses were carried out using EDXS Oxford XMax 20 (Oxford Instruments) spectrometer coupled to the SEM instrument. The spectrometer was calibrated by using the Oxford standards for EDXS analyses. The EDXS results on powder pellets were compared with results of ICP-MS and OES (Optical Emission Spectrometry) analyses of a powder sample from the same ground samples in order to check the accuracy of the EDXS method (Supplementary Materials Table S1).

For pellets and for the surface of cohesive samples, 6 areas were analyzed at ×200 (surface: about 300 µm$^2$). The average and standard deviation were calculated. For the sections, 10 to 13 areas were analyzed along a profile from the edge to the heart of the piece every 300 µm, and at another edge of the piece. The average and standard deviation were also calculated. Three elements were studied in detail: Si, Al, and Fe (major elements in shale and/or ferricrete).

### 3.1.6. PIXE Analyses

PIXE experiments were conducted at the AGLAE (Accélérateur Grand Louvre d'Analyse Elémentaire, Paris, France) facility using a 3 MeV proton extracted beam. The conditions of analyses were chosen according to previous works [67,97]. Because the samples are heterogeneous, they were scanned over a 500 µm$^2$ area. X-ray spectra are recorded by two Si(Li) detectors oriented at 45° to the beam [143]. One is devoted to low energy X-rays (0.1–15 keV) for light elements. The other detector is equipped with selective filters to reduce pileup by attenuating intense X-rays for the detection of heavy elements [144]. We selected the absorber according to the Fe-rich matrix. A 20 µm thick Cr absorber was mounted on the detector. An additional 50 µm thick Al filter was superimposed in order to reduce Cr X-rays induced by the interaction of the primary beam from the sample with the absorber. PIXE spectra were collected between 10 and 15 min in order to obtain more than $2 \times 10^6$ counts. Because of this long acquisition time, only three analyses per samples were done for the powder and the surface and four for the sections.

Elemental concentrations were extracted by using GUPIX [145]. Parameters were calculated by using a DR-N standard and one of the studied samples (14043a; ICP-MS and ICP-OES measurements used as references). The calculated contents in trace elements V, Ni, Cu, and Zn are significantly different from those measured by ICP-MS (relative difference: more than 20%; Supplementary Materials Table S2). These elements were not taken into account in the present study.

### *3.2. Results*
### 3.2.1. XRD Analyses
### Geological

Our results show that the sensitivity of XRD surface analyses on cohesive samples is lower than the one of XRD on oriented powders (Table 5). For shale and phyllite, anatase and potassic feldspars that present a weak peak on powder X-ray patterns are not detected

on the surface. Clay minerals from the kaolinite and chlorite group are not systematically detected when represented as a minor phase and they are not if the main peak is very weak for powder samples (Figure 1A). This is also true for ferricrete pieces. Quartz and clay minerals from the illite and micas group are well detected in cohesive clay mineral matrixes but not on the surface of ferricrete pieces with a cortex.

**Table 5.** Comparison by XRD between non-invasive analyses on cohesive pieces (surface) and invasive analyses on oriented powders (O. powder) for geological samples. Legend—H: hematite; G: goehite; Q: quartz; I/M: illite/mica; K: kaolinite; Ch: chlorite; P: paragonite; Und. CM: undetermined clay mineral (mixed-layer); PF: potassic feldspar; C: calcite; A: anatase.

| Rock Type | Ref. | Mode | H | G | Q | I/M | K | Ch | Pa | Und. CM | PF | C | A |
|---|---|---|---|---|---|---|---|---|---|---|---|---|---|
| Shale | 14042d | O. powder | + | - | +++ | ++ | ++ | nd | nd | nd | - | - | - |
| | | Surface | X | ? | X | X | X | nd | nd | nd | nd | / | nd |
| | 14043a | O. powder | + | - | +++ | ++ | ++ | nd | nd | nd | - | - | - |
| | | Surface | X | ? | X | X | X | nd | nd | nd | nd | X | nd |
| | 14045a | O. powder | + | - | +++ | ++ | ++ | nd | nd | nd | - | - | - |
| | | Surface | X | ? | X | X | X | nd | nd | nd | nd | nd | nd |
| | 14048a | O. powder | ++ | - | +++ | ++ | ++ | nd | nd | nd | - | nd | - |
| | | Surface | X | ? | X | X | X | nd | nd | nd | nd | nd | nd |
| | 14050b | O. powder | ++ | ? | + | +++ | + | nd | nd | nd | - | nd | - |
| | | Surface | X | nd | / | X | X | nd | nd | nd | nd | nd | |
| | 14052a | O. powder | ++ | ? | + | +++ | + | nd | nd | nd | - | nd | - |
| | | Surface | X | nd | / | X | X | nd | nd | nd | nd | nd | nd |
| | 14691b | O. powder | + | nd | +++ | ++ | + | nd | nd | | ++ | - | - |
| | | Surface | X | nd | X | X | nd | nd | nd | nd | X | nd | nd |
| | 14691g | O. powder | + | nd | +++ | ++ | + | nd | nd | | + | - | - |
| | | Surface | X | nd | X | / | / | nd | | | / | | |
| | 14691j | O. powder | + | nd | +++ | ++ | + | nd | nd | nd | + | - | - |
| | | Surface | X | nd | X | X | X | nd | nd | nd | X | nd | nd |
| | 14691k | O. powder | + | nd | +++ | ++ | + | nd | nd | nd | + | - | - |
| | | Surface | X | nd | X | X | nd | nd | nd | nd | nd | nd | nd |
| | 14693d | O. powder | + | nd | +++ | ++ | - | nd | nd | + | - | - | nd |
| | | Surface | + | nd | +++ | + | nd | nd | nd | nd | nd | nd | nd |
| | 14693e | O. powder | + | nd | +++ | ++ | - | nd | nd | + | - | nd | nd |
| | | Surface | X | nd | X | X | nd | nd | nd | nd | nd | nd | nd |
| | 14693g | O. powder | + | nd | +++ | ++ | - | nd | nd | + | - | nd | - |
| | | Surface | - | nd | X | X | nd | nd | nd | nd | nd | nd | nd |
| | 14693j | O. powder | + | nd | +++ | ++ | - | nd | nd | + | - | - | nd |
| | | Surface | X | nd | X | X | nd | nd | nd | nd | nd | nd | nd |
| | 14694e | O. powder | + | nd | +++ | ++ | ? | + | nd | nd | - | nd | nd |
| | | Surface | X | nd | X | X | ? | ? | nd | nd | nd | nd | nd |
| | 14692b | O. powder | + | nd | +++ | ++ | ++ | nd | nd | nd | nd | - | nd |
| | | Surface | X | nd | X | X | X | nd | nd | nd | nd | nd | nd |
| Phyllite (shale group) | 14696c | O. powder | nd | nd | + | +++ | ++ | nd | nd | nd | - | nd | nd |
| | | Surface | nd | nd | X | X | X | nd | nd | nd | - | nd | nd |
| | 14700b | O. powder | nd | nd | + | +++ | + | nd | ? | nd | - | nd | nd |
| | | Surface | nd | nd | X | X | X | nd | ? | nd | nd | nd | nd |
| Indurated shale (ferricrete group) | 14698f | O. powder | + | ++ | +++ | ++ | + | nd | nd | nd | + | nd | nd |
| | | Surface | X | X | X | X | X | nd | nd | nd | X | nd | nd |
| | 14698a (cortex) | O. powder | +++ | nd | ? | + | ++ | nd | nd | nd | - | nd | nd |
| | | Surface | X | nd | ? | nd | nd | nd | nd | nd | nd | nd | nd |
| Ferricrete | 14697a (cortex) | O. powder | nd | +++ | nd | nd | nd | nd | nd | nd | nd | nd | nd |
| | | Surface | nd | X | nd | nd | nd | nd | nd | nd | nd | nd | nd |
| | 14697b (cortex) | O. powder | nd | +++ | + | nd | nd | nd | nd | nd | nd | nd | nd |
| | | Surface | nd | X | nd | nd | nd | nd | nd | nd | nd | nd | nd |
| | 14699a (cortex) | O. powder | nd | +++ | - | nd | + | nd | nd | nd | nd | nd | nd |
| | | Surface | nd | X | nd | nd | nd | nd | nd | nd | nd | nd | nd |
| | 14701a (cortex) | O. powder | nd | +++ | ++ | - | nd | nd | nd | nd | nd | nd | nd |
| | | Surface | nd | X | + | nd | nd | nd | nd | nd | nd | nd | nd |
| | 14701b (cortex) | O. powder | nd | +++ | nd | - | nd | nd | nd | nd | nd | nd | nd |
| | | Surface | nd | X | nd | nd | nd | nd | nd | nd | nd | nd | nd |
| | 14337a (cortex) | O. powder | +++ | nd | + | nd | - | nd | nd | nd | nd | nd | nd |
| | | Surface | X | nd | nd | nd | nd | nd | nd | nd | nd | nd | nd |

Legend—+++: Main phase; ++: secondary major phases (>10%); +: minor phases (<10%); -: a single small peak; nd: not detected; X: detected, several peaks identified; /: detected, a single peak identified.

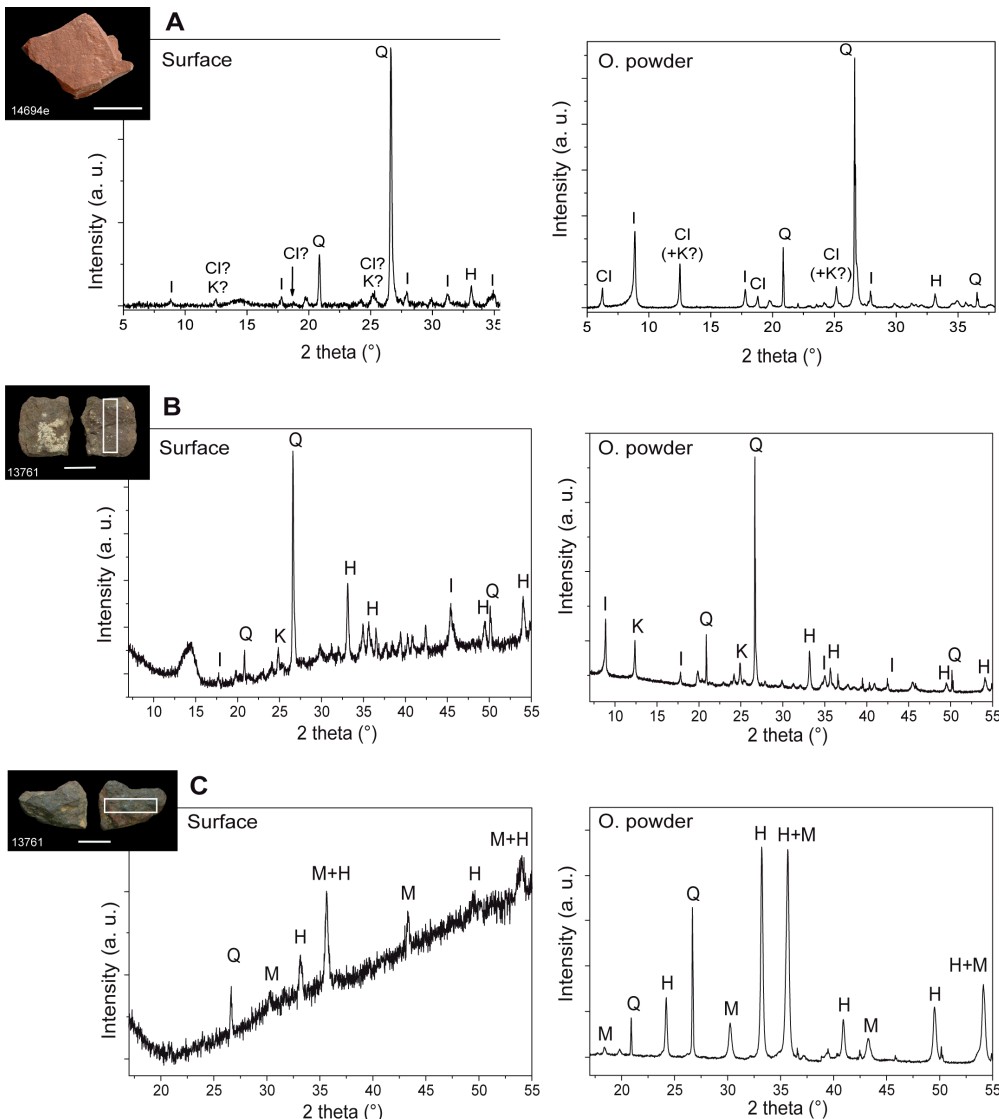

**Figure 1.** XRD patterns of a geological piece of shale (**A**), an archeological piece of shale (**B**), and an archeological piece of ferricrete (**C**): comparison between the surface oriented powder samples (O. powder). Legend—Cl: chlorite; H: hematite; I: illite/mica; K: kaolinite; M: maghemite; Q: quartz; a.u.: arbitrary units.

Archeological

According to the results described above, phases with a single weak peak on the powder X-ray patterns (mainly anatase, potassic feldspar, calcite) were not used to compare archeological samples. When doing so, more than half of the archeological pieces show identical results for phase detection whatever analyzed on the surface or as powder samples. Quartz and kaolinite are more systematically found in powder samples than in cohesive samples for both shale and ferricrete pieces (Table 6; Figure 1B,C). Clay minerals whatever their type are never detected in ferricrete or intermediate raw materials (ferr/shale), especially those exhibiting a cortex (Figure 1C). Interestingly, archeological ferricrete samples which do not show a highly developed cortex do not contain clay minerals at all (below the detection limit of powder oriented XRD analyses).

In certain shale and ferricrete pieces, maghemite is detected on the surface of the samples but not in the powder. Hematite is not detected on the surface of two samples despite it is present in the powder; the main iron oxide observed in both the surface and

the powder is maghemite (13716; 13777). Goethite is not detected on the surface of another sample despite being detected in the powder, hematite being identified instead (13722).

**Table 6.** Comparison between non-invasive analyses on cohesive pieces and invasive analyses on oriented powders by XRD for archeological samples. Legend—*: proxy for geological origin; in bold: main phase detected.

| Rock Type | Ref. | Oriented Powder | Surface | Difference of Composition |
|---|---|---|---|---|
| Shale/sandstone | 13716 | Quartz, I/M, maghemite, hematite | Quartz, maghemite | Heat, sensitivity |
| Shale | 13681 | Quartz, I/M, kaolinite, hematite | Identical | - |
| Shale | 13711 | I/M, quartz, hematite, kaolinite | Identical | - |
| Shale | 13715 | Quartz, I/M, kaolinite, hematite | Identical | - |
| Phyllite | 13718 | I/M, hematite | Identical | - |
| Shale | 13723 | Hematite, I/M, pyrophyllite * | Identical * | - |
| Shale | 13752 | I/M, quartz, hematite, goethite, kaolinite, anatase | Quartz, I/M, hematite, goethite | Sensitivity |
| Shale | 13757 | I/M, kaolinite, quartz, hematite, anatase | Kaolinite, I/M, hematite | Sensitivity, patina? |
| Shale | 13758 | Quartz, kaolinite, I/M, hematite | Identical | - |
| Shale | 13761 | I/M, quartz, hematite, kaolinite | Identical | - |
| Shale | 13771 | I/M, quartz, hematite | I/M, hematite | Patina? |
| Shale | 13773 | Quartz, hematite, I/M, kaolinite | Identical | - |
| Shale | 13777 | Quartz, kaolinite, I/M, hematite | I/M, maghemite | Heat, sensitivity, patina? |
| Shale | 13783 | I/M, quartz, hematite, kaolinite | Identical | - |
| Shale | 13804 | Quartz, I/M, hematite, kaolinite | Identical | - |
| Ferr/shale | 13725 | Hematite | Hematite, quartz (maghemite?) | Cortex |
| Ferr/shale | 13727 | Maghemite, hematite, quartz | Identical | - |
| Ferr/shale | 13750 | Hematite, I/M, kaolinite | Hematite, quartz | Sensitivity, cortex |
| Ferricrete | 13712 | Hematite | Identical | - |
| Ferricrete | 13690 | Maghemite, hematite, quartz | Identical | - |
| Ferricrete | 13722 | Goethite, kaolinite, hematite, quartz | Hematite | Heat, sensitivity, cortex |
| Ferricrete | 13724 | Hematite, quartz | Maghemite, hematite, quartz | Heat |
| Ferricrete | 13741 | Hematite, quartz | Identical | - |
| Ferricrete | 13743 | Hematite, maghemite | Hematite | Heat |
| Ferricrete | 13749 | Hematite, maghemite | Identical | - |
| Ferricrete | 13762 | Hematite, quartz, kaolinite | Quartz, hematite | Sensitivity, cortex |
| Ferricrete | 13793 | Hematite, maghemite, quartz | Maghemite, hematite | Cortex |

### 3.2.2. SEM-EDXS Analyses
Geological

When powder and cohesive geological samples are compared, we observe a relatively good adequacy for four samples (three shale and one ferricrete), and great discrepancy for one ferricrete in Si, Al, and Fe contents (Table 7; Figure 2). The latter is composed of a crypo- to microcrystalline matrix of iron oxide in which coarse quartz grains are heterogeneously dispersed. The area of analysis is too small to overcome this great textural heterogeneity. The texture of the other samples is more homogenous under the scale of analysis chosen (Figure 3). The relative difference between the two averages (surface and powder) is lower than 10% for three of them and lower than 20% for the last one. This difference remains acceptable and could probably be improved by increasing the number of analyses for cohesive samples.

**Table 7.** Comparison by EDXS between non-invasive analyses on cohesive pieces (surface) and invasive analyses on powder pellets (powder) for geological samples. Results were normalized to 100% by using a conventional list of elements (ICP-OES data). Legend—SD: standard deviation.

| Sample | Mode | Measurement | Na$_2$O | MgO | Al$_2$O$_3$ | SiO$_2$ | P$_2$O$_5$ | K$_2$O | CaO | TiO$_2$ | MnO | Fe$_2$O$_3$ | Total |
|---|---|---|---|---|---|---|---|---|---|---|---|---|---|
| 14043a | Surface | AVERAGE | 0.5 | 0.7 | 24.2 | 56.5 | 0.7 | 5.1 | nd | 1.2 | nd | 11.0 | 100.0 |
| | | SD | 0.0 | 0.1 | 0.5 | 1.2 | 0.1 | 0.2 | - | 0.1 | - | 1.1 | - |
| | Powder | AVERAGE | 0.5 | 0.9 | 25.0 | 55.9 | 0.4 | 5.0 | 0.1 | 1.1 | 0.1 | 11.0 | 100.0 |
| | | SD | 0.1 | 0.0 | 0.2 | 0.6 | 0.1 | 0.0 | 0.0 | 0.1 | 0.1 | 0.3 | - |
| 14050b | Surface | AVERAGE | 0.3 | 1.0 | 25.2 | 47.9 | 0.7 | 6.4 | 0.1 | 1.4 | 0.1 | 16.9 | 100.0 |
| | | SD | 0.0 | 0.1 | 0.9 | 1.3 | 0.2 | 0.3 | 0.1 | 0.2 | 0.1 | 1.5 | - |
| | Powder | AVERAGE | 0.8 | 1.2 | 26.2 | 46.5 | 0.5 | 6.3 | 0.3 | 1.3 | 0.1 | 16.7 | 100.0 |
| | | SD | 0.0 | 0.0 | 0.1 | 0.1 | 0.1 | 0.0 | 0.0 | 0.1 | 0.0 | 0.1 | - |
| 14696c | Surface | AVERAGE | 7.2 | 2.5 | 24.0 | 52.4 | nd | 4.8 | nd | 1.1 | nd | 8.0 | 100.0 |
| | | SD | 0.8 | 0.1 | 1.9 | 1.0 | - | 0.4 | - | 0.1 | - | 0.4 | - |
| | Powder | AVERAGE | 2.4 | 2.1 | 27.0 | 55.8 | nd | 5.7 | 0.1 | 1.0 | nd | 6.0 | 100.0 |
| | | SD | 0.3 | 0.2 | 0.3 | 0.3 | - | 0.1 | 0.1 | 0.1 | - | 0.2 | - |

**Table 7.** *Cont.*

| Sample | Mode | Measurement | Na₂O | MgO | Al₂O₃ | SiO₂ | P₂O₅ | K₂O | CaO | TiO₂ | MnO | Fe₂O₃ | Total |
|---|---|---|---|---|---|---|---|---|---|---|---|---|---|
| 14697a | Surface | AVERAGE | 1.4 | 0.4 | 3.9 | 2.9 | 1.9 | 0.1 | 0.1 | 0.1 | nd | 89.3 | 100.0 |
| | | SD | 1.3 | 0.1 | 1.1 | 0.6 | 0.2 | 0.1 | 0.1 | 0.2 | - | 3.3 | - |
| | Powder | AVERAGE | 0.4 | 0.5 | 4.0 | 2.5 | 2.5 | 0.1 | 0.2 | nd | nd | 89.7 | 100.0 |
| | | SD | 0.0 | 0.1 | 0.4 | 0.1 | 0.1 | 0.0 | 0.0 | - | - | 0.4 | - |
| 14699a | Surface | AVERAGE | 0.1 | 0.1 | 8.2 | 6.3 | 3.9 | 0.2 | 0.1 | nd | nd | 81.0 | 100.0 |
| | | SD | 0.1 | 0.1 | 5.3 | 8.0 | 1.0 | 0.2 | 0.1 | - | - | 12.7 | - |
| | Powder | AVERAGE | 0.2 | 0.2 | 17.7 | 17.7 | 3.3 | 0.4 | nd | 0.1 | nd | 60.4 | 100.0 |
| | | SD | 0.0 | 0.0 | 0.2 | 0.2 | 0.1 | 0.0 | - | 0.1 | - | 0.4 | - |

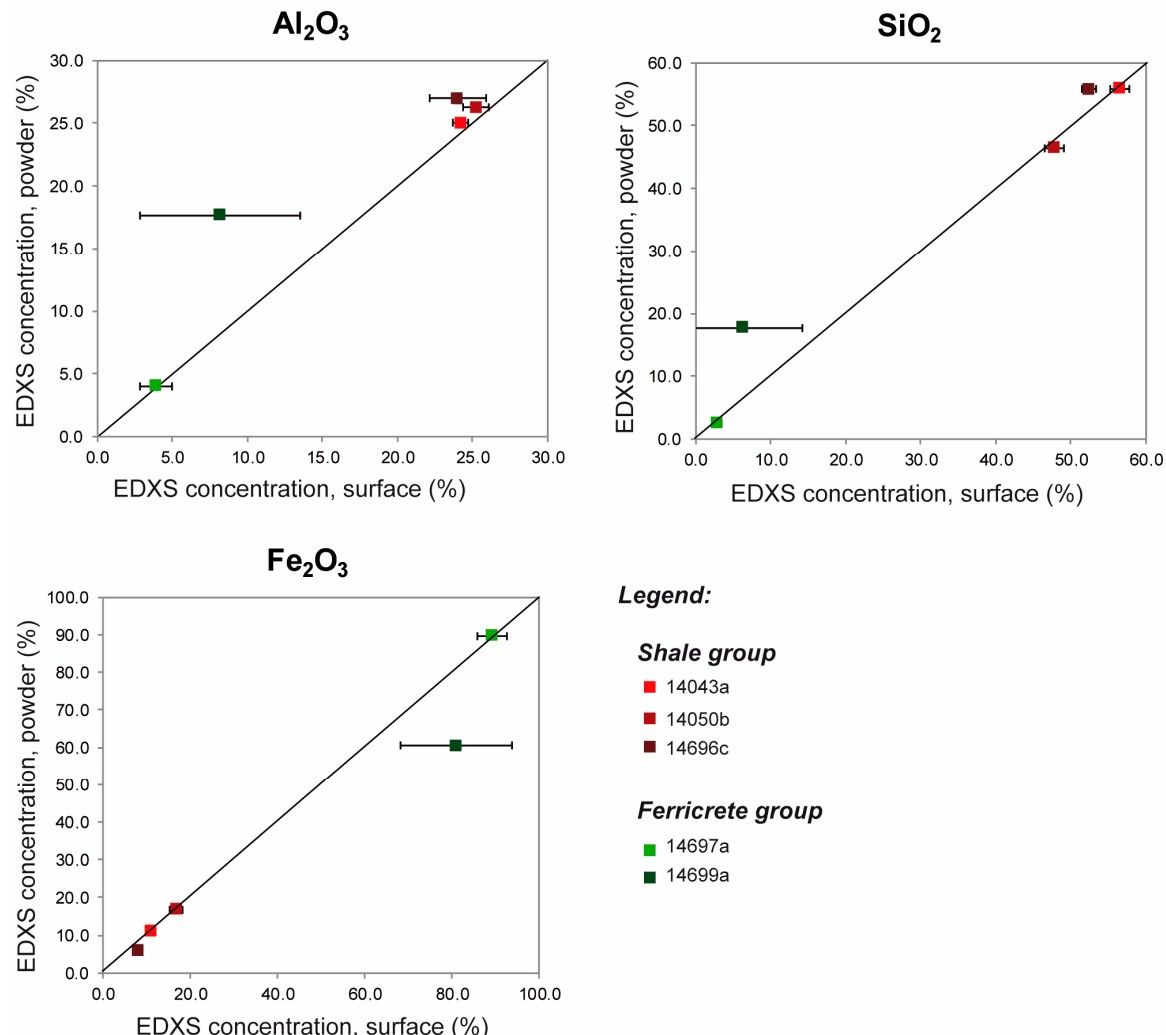

**Figure 2.** Results of EDXS analyses of cohesive geological pieces (surface) compared with the analysis of pellets (powder) for major elements Al, Si, and Fe.

Archeological

The comparison between the measurements on the surface and the section shows that small quantities of soluble salts remain at the surface. Elements Na, Mg, P, and Ca are in higher concentration at the surface relative to the section of the ferricrete and shale pieces (<1.5% in mass oxide; Figure 4; Supplementary Materials Table S3). They form a surficial layer of water soluble minerals. The concentration in Si is significantly lower on the surface of three samples, the two shale pieces (13681 and 13773) and one ferricrete piece (13690) (Figure 4). The absolute difference is of 9% to 16%. In two ferricrete pieces, Fe content is a bit lower on the surface (13712, 13741) while it is a bit higher in one shale piece (13681), and Al content is higher in the other shale piece (13773). The high decrease in Si

content at the surface of the samples can be interpreted as the presence of a thin coating of quartz grains. In ferricrete pieces, the formation of a patina or a cortex might lead to slight Fe impoverishment.

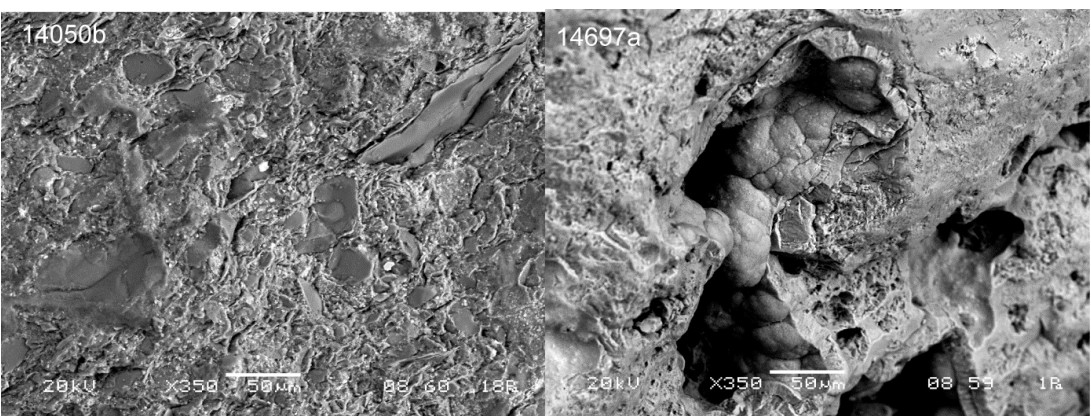

**Figure 3.** SEM images (BSE, back scattered electron mode) of the surface of two geological pieces studied, showing the coarse inclusions in one shale (14050b) and the matrix of iron oxide crystals in one ferricrete (14697a).

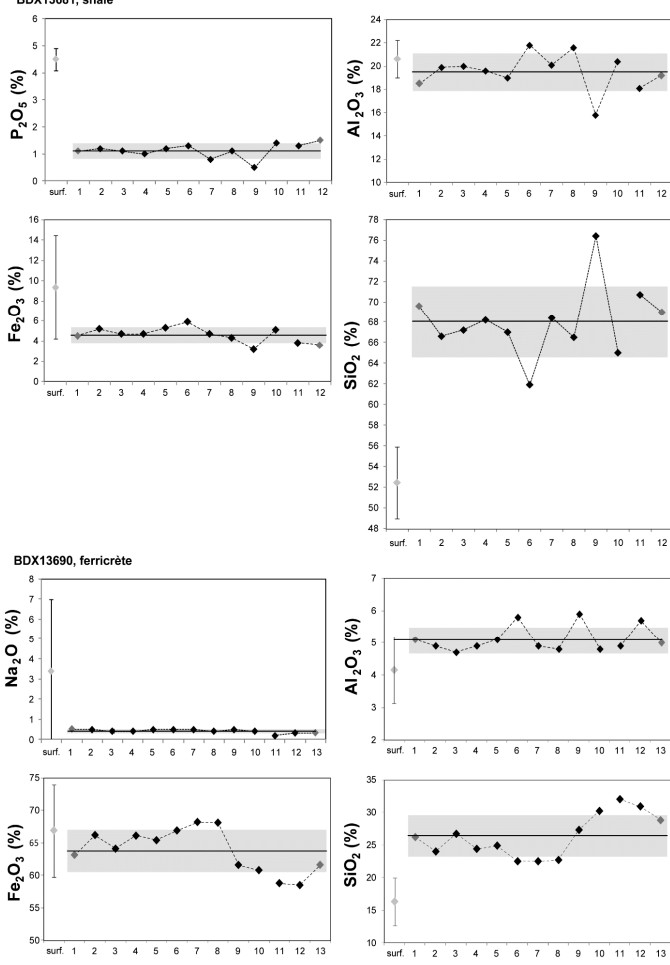

**Figure 4.** Main EDXS results of two archeological pieces, a shale (13681) and a ferricrete (13690): comparison between the surface and the section. Diamonds in light gray: average with standard deviation calculated from six measures on the surface. Diamonds in dark gray: edges of the section. Diamonds in black: inside of the section. The black line represents the average from the measurements on the section and the gray area the standard deviation. Legend—Surf.: Surface.

### 3.2.3. PIXE

Geological

The adequacy between powder and surface analyses for geological samples is lower for PIXE analyses than for EDXS analyses. It is mostly Al and Si contents that are affected (Figure 5; Table 8). The surface of a shale sample (14696c) is noticeably enriched in Na and Cl, explaining the discrepancies observed. For the rest of the samples, the absence of compositional pattern suggests that the number of analyses carried out at the surface was too low to overcome heterogeneity problems. With regard to trace elements, a good adequacy between the surface and pellets is observed for Ga, As, Rb, Y, Nb, Pb, and Th: the $R^2$ is systematically higher than 0.95 (Figure 5).

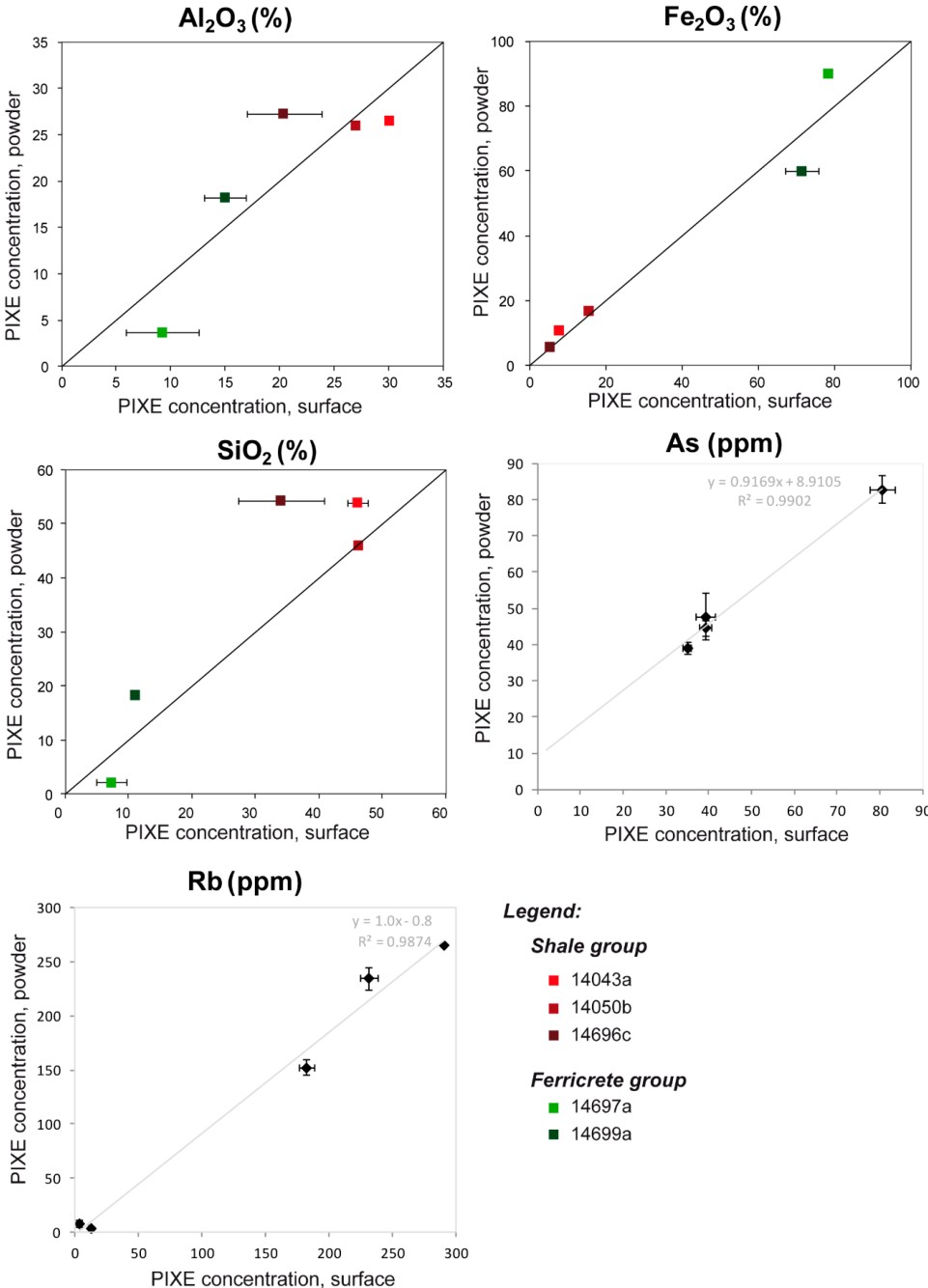

**Figure 5.** Results of PIXE analyses of cohesive geological pieces (surface) compared with the analysis of pellets (powder) for major elements Al, Si, and Fe, as well as trace elements Rb and As.

**Table 8.** Comparison by PIXE between non-invasive analyses on cohesive pieces (surface) and invasive analyses on powder pellets (powder) for geological samples. Results were normalized to 100% by using all elements quantified by PIXE. Legend—nd: not detected.

| Sample | Mode | Maesurement | Na2O % | MgO % | Al2O3 % | SiO2 % | P2O5 % | SO3 % | Cl % | K2O % | CaO % | TiO2 % | MnO % | Fe2O3 % | Cr ppm | Ga ppm | As ppm | Rb ppm | Sr ppm | Y ppm | Zr ppm | Nb ppm | Ba ppm | Pb ppm | Th ppm |
|---|---|---|---|---|---|---|---|---|---|---|---|---|---|---|---|---|---|---|---|---|---|---|---|---|---|
| 14043a | Surface | AVERAGE | 1.6 | 0.5 | 30.1 | 46.0 | 4.5 | 1.3 | 2.4 | 4.0 | 0.2 | 1.0 | 0.05 | 7.7 | 64 | 26 | 48 | 152 | 290 | 48 | 394 | 22 | 443 | 58 | 23 |
|  |  | SD | 0.3 | 0.0 | 0.3 | 1.6 | 0.6 | 0.1 | 0.1 | 0.1 | 0.0 | 0.2 | 0.00 | 1.1 | 5 | 4 | 6 | 26 | 39 | 7 | 28 | 8 | 215 | 13 | 3 |
|  | Powder | AVERAGE | 0.4 | 0.7 | 26.5 | 53.9 | 0.3 | 0.1 | 0.3 | 4.9 | 0.1 | 1.4 | 0.06 | 10.4 | 71 | 32 | 39 | 183 | 147 | 50 | 439 | 25 | 660 | 50 | 22 |
|  |  | SD | 0.0 | 0.0 | 0.2 | 0.2 | 0.0 | 0.0 | 0.0 | 0.1 | 0.0 | 0.0 | 0.01 | 0.4 | 19 | 2 | 2 | 3 | 7 | 6 | 107 | 3 | 155 | 4 | 2 |
| 14050b | Surface | AVERAGE | 0.2 | 0.8 | 27.0 | 46.2 | 0.4 | 0.9 | 0.1 | 6.1 | 0.1 | 1.8 | 0.09 | 15.5 | 155 | 38 | 83 | 234 | 204 | 76 | 351 | 33 | 669 | 68 | 32 |
|  |  | SD | 0.0 | 0.0 | 0.3 | 0.5 | 0.0 | 0.3 | 0.0 | 0.2 | 0.0 | 0.1 | 0.01 | 0.2 | 6 | 0 | 4 | 9 | 13 | 11 | 36 | 3 | 106 | 2 | 2 |
|  | Powder | AVERAGE | 0.5 | 0.9 | 26.0 | 46.0 | 0.3 | 0.3 | 0.5 | 6.0 | 0.3 | 1.7 | 0.09 | 16.5 | 130 | 37 | 81 | 232 | 201 | 74 | 334 | 34 | 608 | 67 | 27 |
|  |  | SD | 0.1 | 0.0 | 0.1 | 0.3 | 0.0 | 0.0 | 0.0 | 0.0 | 0.0 | 0.0 | 0.00 | 0.2 | 32 | 2 | 3 | 3 | 3 | 7 | 75 | 4 | 246 | 5 | 4 |
| 14696c | Surface | AVERAGE | 13.5 | 2.4 | 20.3 | 33.8 | 0.2 | 3.2 | 18.1 | 4.1 | 0.1 | 1.0 | nd | 4.7 | 67 | 24 | nd | 265 | 75 | 31 | 327 | 19 | 592 | 9 | 30 |
|  |  | SD | 6.7 | 0.5 | 3.4 | 6.7 | 0.0 | 0.0 | 6.2 | 0.5 | 0.0 | 0.2 | - | 0.2 | 33 | 1 | - | 15 | 9 | 1 | 301 | 4 | 118 | 3 | 7 |
|  | Powder | AVERAGE | 1.9 | 1.5 | 27.3 | 54.3 | 0.0 | 0.2 | 3.1 | 5.3 | 0.1 | 1.0 | 0.01 | 4.8 | 109 | 27 | 2 | 291 | 70 | 37 | 204 | 17 | 599 | 9 | 25 |
|  |  | SD | 0.1 | 0.1 | 0.2 | 0.1 | 0.0 | 0.0 | 0.1 | 0.2 | 0.0 | 0.0 | 0.00 | 0.2 | 53 | 1 | - | - | 7 | 13 | 60 | 2 | 174 | 1 | 6 |
| 14697a | Surface | AVERAGE | 0.8 | 0.5 | 9.2 | 7.2 | 1.6 | 0.7 | 0.4 | 0.3 | 0.1 | 0.1 | nd | 78.3 | nd | 5 | 39 | 8 | 378 | 4 | 23 | 2 | 3068 | 12 | 11 |
|  |  | SD | 0.1 | 0.1 | 1.9 | 2.2 | 0.2 | 0.1 | 0.1 | 0.1 | 0.0 | 0.0 | - | - | - | 1 | 2 | 3 | 8 | 3 | 6 | 0 | 456 | 1 | 4 |
|  | Powder | AVERAGE | 0.3 | 0.3 | 3.7 | 2.2 | 1.8 | 0.5 | 0.5 | 0.1 | 0.2 | nd | nd | 90.1 | 59 | 5 | 35 | 4 | 55 | 5 | 18 | 3 | 441 | nd | 8 |
|  |  | SD | 0.0 | 0.0 | 0.1 | 0.1 | 0.0 | 0.0 | 0.0 | 0.0 | 0.0 | - | - | 0.6 | 37 | - | 1 | 2 | 4 | 2 | 1 | 1 | 140 | - | 1 |
| 14699a | Surface | AVERAGE | 0.2 | 0.3 | 15.0 | 11.0 | 1.4 | 0.2 | 0.0 | 0.2 | 0.1 | 0.2 | nd | 71.4 | 117 | 15 | 45 | 4 | 13 | 9 | 32 | 2 | 86 | 30 | 6 |
|  |  | SD | 0.0 | 0.0 | 0.1 | 0.4 | 0.0 | 0.0 | 0.0 | 0.0 | 0.0 | 0.0 | - | 0.4 | 47 | 1 | 2 | 1 | 1 | 1 | 3 | - | 40 | 1 | - |
|  | Powder | AVERAGE | 0.1 | 0.2 | 18.2 | 18.3 | 2.4 | 0.1 | 0.0 | 0.4 | 0.0 | 0.1 | nd | 59.9 | 98 | 5 | 39 | 13 | 8 | 6 | 16 | 2 | 79 | 25 | nd |
|  |  | SD | 0.0 | 0.0 | 0.2 | 0.3 | 0.1 | 0.0 | 0.0 | 0.0 | 0.0 | 0.0 | - | 0.5 | 31 | 1 | 2 | 2 | 2 | 2 | 4 | - | - | 4 | - |

Archeological

The analyses of five sections confirms that the surface of the pieces is enriched in Na, Mg, P, and Ca (Supplementary Materials Table S4). In addition, Si concentrations are significantly much lower in all shale or shale/sandstone pieces (13681; 13715; 13716; 13773; Figure 6A) and it shows variations in the inside of the ferricrete (13741). Al contents are a bit higher in two shale pieces (13681; 13773). This is consistent with the main tendency observed by EDXS and allows to refine some of the hypotheses proposed above: quartz grains might be covered by a thin coating, sometimes enriched in Al; coarse quartz grains are heterogeneously distributed in certain ferricrete pieces.

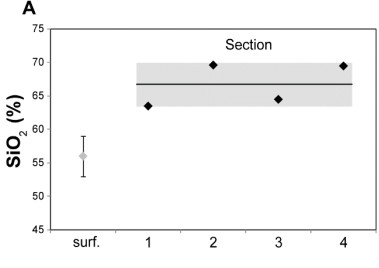
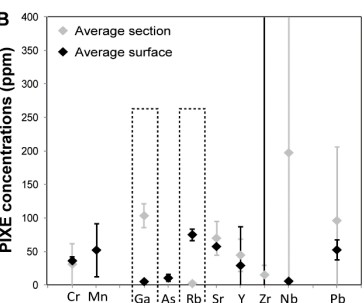

**Figure 6.** Main PIXE results for a piece of shale/sandstone (13716): comparison between the surface and the section. (**A**) Results for the major element Si. Diamonds in light gray: average with standard deviation calculated from three measures on the surface. Diamonds in black: inside of the section. The black line represents the average from the measurements on the section and the gray area the standard deviation. (**B**) Results for trace elements (average and standard deviation only). Legend— Surf.: Surface.

Tendencies in trace element contents are very variable depending on the trace element and the raw material considered. For shale pieces, the distribution of Zr and Ba is heterogeneous (high standard deviation; Figure 6B). This may explain the low adequacy between powder and surface in geological samples. The adequacy between the surface and the section is quite good in all samples for Cr, As, Y, Nb, Pb, Th, and except in one sample (13716) for Ga and Rb (Figure 6B).

### 3.3. Discussion

3.3.1. Instrumental Limits

Under the analysis conditions followed in this study, XRD on cohesive samples clearly appears to be less sensitive. Only phases with more than one diffraction peak on powder XRD patterns and phases whose identification is not much dependent on crystal orientation are systematically detected in cohesive samples. The area analyzed when using a conventional diffractometer with a Göbel mirror is sufficiently large as to adequately sample a specimen's heterogeneity in a representative manner. On the contrary, the heterogeneous nature of cohesive samples with coarse inclusions does tend to introduce biases when elemental analyses are carried out on areas smaller than 500 mm$^2$. Differences between cohesive and powder samples are lower in EDXS than in PIXE analyses, and this likely relates to the higher number of analyses conducted on each cohesive piece. To summarize, non-invasive XRD analyses in these conditions are less sensitive but still accurate for ochre characterization, while non-invasive EDXS and PIXE analyses require multiple measurements (more than six in our case) in order to overcome the biases introduced by ochre grain-size and mineralogical heterogeneity.

3.3.2. Post-Depositional Alterations

The first surficial post-depositional modification identified from our results was the presence of deposits on the surface of several samples, despite the fact that the surface had been washed. However, the elemental and mineralogical composition of these superficial salts cannot be mistaken with the internal composition of both the shale and ferricrete pieces given they are not present within these two geomaterials. Two other patterns in our results can also be related to post-depositional alterations: (1) kaolinite and quartz are less frequently detected in archeological samples relative to geological samples; (2) Si content is much lower on the surface of all archeological shale pieces (both EDXS and PIXE analyses). This could be due to the formation of a thin coating with a distinct composition on the surface of pieces that have a matrix constituted by clay minerals (shale, phyllite, shale/sandstone), which would potentially cover quartz grains. This layer may have formed within the archeological deposits but may also be a feature inherited from the sub-primary deposits they come from, meaning that already altered pieces may have been introduced into the site. In order to determine the validity of both of these hypotheses, further data would need to be obtained via the analysis of geological shale pieces with a patina. For ferricrete pieces, the presence of a weathering cortex influences the results of XRD surface analyses for both geological and archeological samples, while this same weathering cortex would appear to affect the EDXS results only for archaeological samples. This may indicate that the sub-primary geological cortex has elemental composition that is more or less similar to the center of the piece, albeit with lower degrees of crystallinity. With regard to archeological ferricrete pieces, post-depositional processes likely induced the migration of a part of the elements at the surface of the sub-primary cortex, as was the case for most archeological pieces.

Surprisingly, trace elements seem to be less affected than major elements by alteration processes. We observed differences in trace element contents at the surface of one ochre piece only. The number of trace elements accurately determined by PIXE was, however, quite low in our case (nine elements). Further research is necessary in order to evaluate the role of trace element migration in the formation of secondary patinas in archaological deposits from Southern African rock shelters, such as at Diepkloof rock shelter.

Additionally, post-depositional processes likely contributed to the differences in iron oxide and oxi-hydroxide identified on the surface of archeological pieces relative to powders. Such differences could stem from one of two effects, either there are differences in detection sensitivity, or mineralogical compositions may truly be distinct. We are inclined to favor this second hypothesis for the following reasons. When differences are perceived, the iron oxides or oxy-hydroxides detected in powders always correspond to lower heat-intensities relative to surface readings (goethite in powder and hematite on sample surface,

maghemite on sample surfaces and hematite in powder). Furthermore, fire features are frequent all over the site sequence [146], and the heat treatment of silcrete has been identified in several stratigraphic units [147,148]. The high quantity of intentional fire use on the site seems more parsimonious therefore with the second of the above-mentioned hypotheses.

### 3.3.3. Consequences for Archeological Inferences

Having now determined both the potential instrumental limits and possible post-depositional alterations, we can now move forward in the construction of a rigorous interpretive framework for our results. Firstly, the two main rock categories observed (shale, including phyllite, and ferricrete) remain clearly distinguishable by their percentages in major elements and by their mineralogical compositions. Interestingly, the absence of clay minerals in cohesive ferricrete pieces allows for consistent identification of ferricrete relative to shale pieces with a clay mineral matrix. However, this particular collection may represent an isolated case study given the salty nature of the superficial coatings of post-depositional origin identified on the archeological pieces and the absence of similar compounds in their internal composition. By comparison, when archeological deposits are rich in clay minerals, sediment adhering to archeological remains may sometimes be difficult to distinguish from the actual surface of a piece of mudstone or shale.

Regarding the identification of geological and geographical origins, it would appear that non-invasive XRD analyses are much more limited seeing as they are unable to detect all families of clay minerals in cohesive samples. Moreover, we cannot use the criteria of illite crystallinity as proposed for powder samples, because the conditions of analyses are not suitable for peak deconvolution [17]. Pyrophyllite, a major proxy for a non-local geological origin was nonetheless clearly detected in one ochre piece. Qualitative SEM-EDXS results can also be used. The micro-structure of the ochre samples allows us to distinguish between true shale (*sensu stricto*), phyllite, and ferricrete, and these are relevant distinctions for determining particular geological origins [28].

Finally, we showed that the contents of six trace elements are not affected by elemental migrations at the surface of archeological samples, among which five are accurately quantified on the surface of geological samples in comparison with homogenized powders: As, Y, Nb, Pb, and Th. This list is quite different from the list of trace elements quantified by ICP-MS that were used to discriminate the geological shale samples, which were As, Ba, Cr, Sb, and V [28]. Moreover, it is quite a low number of trace elements for the discrimination of geological ochre sources in general [18,19,38,39,56,58–61,63,64,85,122]. Nonetheless, the discrimination between two sources based on a few trace elements is possible as soon as these sources are characterized by consistent and significant differences. This might be the case, for instance, if they come from different types of geological formations, as shown by a previous PIXE study [36]. It is also possible to carry out multi-variate analyses on this set of variables in order make an efficient selection of the trace elements that have the highest weights in inter-source variations. After a careful consideration of the limitations of the PIXE analyses conducted at the surface of heterogeneous ochre pieces (low representativeness of small area PIXE surface analyses for instance), and an evaluation of post-depositional aspects, we can now securely conclude that provenance research using non-invasive PIXE analyses are theoretically possible on this particular collection. Further investigations will be carried out on a larger selection of pieces in the future.

As a whole, we demonstrated that non-invasive analyses can be used to address provenance issues in the Diepkloof ochre collection, though they do not allow for specific attributions to a particular source. Moreover, such methods do not allow determining the geological origin of each single archeological piece. The methodology we employed here in order to enhance the robustness of our inferences should be extended to other contexts when analysts are limited to sacrificing only small numbers of samples for more invasive methods of analyses.

Although evaluating the efficiency of non-invasive analyses to detect the evidence of heat treatment was not part of the initial goals of this project, we did find evidence that

ochre pieces were heated by using non-invasive XRD analyses. The differences observed between the inside and the surface of the analyzed samples favor incidental burning rather than intentional heating [82]. These results were nevertheless very useful for discussions regarding the possibility of heat-treated ochre pieces at the Klasies river main site, as these were analyzed in similar conditions [27].

## 4. Conclusions

A thorough review of both invasive and non-invasive methods used in ochre studies has shown the degree to which these different approaches can complement one-another. While conventional invasive methods provide more robust and consistent results, some non-invasive or minimally invasive studies do successfully address some aspects of raw material transformation and provenience issues. Our results provide new information on the relevance of non-invasive XRD, SEM-EDXS, and PIXE methods used in ochre characterization. The methodology we followed, which was based on the comparison of invasive and non-invasive analyses of both geological and archeological samples, was particularly useful for demonstrating the respective limitations of these methods and the potential effects of post-depositional processes on the surfaces of archeological specimens. Once these biases were clearly identified, they were considered and were used to temper the final results and the resulting inferences. While these were ultimately less precise due to the aforementioned biases, they nevertheless provided accurate and relevant information on raw material characterization, their provenance, and the question of heat treatment. On the whole, non-invasive XRD analyses appear to be a key method for addressing a broad range of archeological issues with regard to ochre characterization. The qualitative and semi-quantitative results of SEM-EDXS surface analyses were complementary to XRD analyses by adding information on the micro-structure of the pieces and the ranges of composition in major elements.

The methodological results presented here, when combined with other previously published results, provide clear future avenues with regard to how both robust invasive and non-invasive analyses can be combined in order to respond to different lines of archeological questioning. Nonetheless, the relative success of non-invasive methods for the characterization of ochre materials shall not move us away from other avenues of in-depth methodological research as initiated in this work. In particular, taphonomic biases require a more general and broad scale evaluation; minimally invasive methods such as LA-ICP-MS may represent a more optimal compromise relative to non-invasive PIXE analyses given that they are affected by these biases and by instrumental limits related to the analysis of iron-rich matrixes. These represent starting points for methodological improvements in archeological ochre studies at a broader scale.

**Supplementary Materials:** The following are available online at https://www.mdpi.com/2075-163X/11/2/210/s1, Table S1: EDXS data compared with ICP-OES (major) data for 5 geologic samples (powder pellets), Table S2: Comparison of PIXE data with ICP-OES (major) and ICP-MS (traces) results for 5 geologic samples (powder pellets), Table S3: Detailed results of SEM-EDXS analyses of archeological samples (surface and section), Table S4: Detailed results of PIXE analyses of archaeological samples (surface and section).

**Funding:** Région Aquitaine, France: Prehistoric ochre project.

**Institutional Review Board Statement:** Not applicable.

**Informed Consent Statement:** Not applicable.

**Data Availability Statement:** Not applicable.

**Acknowledgments:** The author wishes to thank Floréal Danieal (IRAMAT-CRP2A), Pierre-Jean Texier, and Pierre Guibert for the supervision of the PhD thesis from which part of the results presented here come from. Thank you to Stéphane Dubernet and Nadia Cantin for their advice on XRD acquisition; to Yannick Lefrais for his advice in SEM-EDXS analyses; to Lucile Beck, Laurent Pichon and Claire Pacheco (C2RMF) for their advice and help in PIXE analyses at the AGLAE platform; and

to John Parkington and Judith Sealy (University of Cape Town) for their help with the export permit application. We are also grateful to the French Ministry of Foreign Affairs (MAE), the Aquitaine region, the Provence-Alpes-Côte-d'Azur region, and the Centre National de la Recherche Scientifique (CNRS) for their financial support of the Diepkloof excavations. This ochre characterzsation project was funded by the University of Bordeaux 3, the Centre National de la Recherche Scientifique, and the Région Aquitaine, France.

**Conflicts of Interest:** The author declares no conflict of interest.

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
