# Peer review of "Invasive and Non-Invasive Analyses of Ochre and Iron-Based Pigment Raw Materials: A Methodological Perspective"

_minerals, doi:10.3390/min11020210_

Round 1

Reviewer 1 Report

The manuscript offers two important contributions to the specialist field of ochre/iron oxide characterization and provenance research: (1) a thorough and detailed overview of key issues related to different approaches to iron oxide characterization, and, (2) a demonstration of the impacts using of different methods (and potential inferences based on their results) using data from Dayet’s ongoing research at Diepkloof rockshelter.

The first half of the manuscript presents a detailed overview of invasive (NAA, ICP-MS, other ‘bulk’ analyses) vs. non-invasive methods (XRF, PIXE, XRD, the latter of which as demonstrated can produce different results depending on how samples are prepared). Dayet has combed through the last two decades of ochre characterization literature and summarized, in a clearly organized table, methods and preparation techniques used by the array of researchers. This table, and the accompanying bibliography, is a great quick-reference resource for scholars in our area of interest. The literature review is thorough, highlights salient points, and will be a great tool for emerging scholars navigating the literature and seeking to make informed decisions about their own ochre research design. It is expertly written by someone who has clearly invested significant time and focus on the issues of multi-method approaches and taphonomic impacts to ochre assemblages. The technical language and descriptions are appropriately scoped for this journal and the intended readership. I will be happy to cite this and to share it with my students. 

The first half of the manuscript sets the contextual stage for the second half where Dayet shows the reader an applied example. The results are quite clear. For instance, they have shown that some elements suffer from higher degree of inter-sample/spatial variation (which is not a new revelation, but one of the first attempts to go to the trouble of demonstrating it systematically). They also show how and where surface vs interior measurements vary, and where presence of cortex or patination (salt formation, clay deposition, and other taphonomic impacts) can influence the results of compositional analyses. These key issues should have  significant implications for researchers who are assessing their archaeological assemblage, considering which method(s) to employ, and how to subsequently prepare their research materials and set up their experimental conditions (e.g. cleaning, burring, grinding, pressing pellets, analytical ‘spot size’, how many assays/data points to collect for a representative signal, etc.). Every step in the decision-making process can introduce different types, and intensities, of bias. I think this is shown quite clearly (and in the most impactful way, in my mind) with particular regard to the inability (lack of sensitivity) to analyze certain clay minerals at low-angle XRD, which could be viewed as a false-negative in terms of provenance inferences. This is an issue that I’ve intuited and encountered in my own research and have been trying to determine: how deep (in cross section) do taphonomic impacts appear in an artifact specimen? Does surface cleaning and burring of tens of microns sufficiently reduce the potential impacts of taphonomy? Of course, this depends on the microstructure and porosity of the mineral matrix, but as Dayet likely knows, this issue presents quite a practical challenge for analyzing large and highly variable assemblages. 

What I think could be addressed in a bit more detail is how the differences in the methods may or may not have impacted Dayet’s own previous interpretations. In their earlier papers on the Diepkloof ochre assemblage, the results suggested some variation in the ochre provisioning, with at least some of the assemblage attributed to non-local shale. Have the new PIXE data (which wasn’t included in the 2013 paper), or any new insights developed since then, compelled Dayet to reinterpret the earlier data or reconsider any provenance assignments? If any new assessments (or, even a positive reinforcement of the previous interpretations) have come to light with the new data, this would be a key point of discussion to interested readers. This is mentioned briefly in the conclusion, but mostly from a methodological standpoint and not an interpretive one. I think a stronger link to the earlier hypotheses posed in the previous publication could help drive the point home: how, in this case study, could the use of different methods impact the interpretation of ochre provisioning activities at Diepkloof? If the data patterns might have been interpreted differently in light of new or different information, this is important to know. If the data patterns are consistent and reinforce previous inferences, that would be equally important to highlight. If I had to suggest any additional content to add to the manuscript, a clearer and more detailed link to the previously published data that highlights any new interpretations would be a highly valuable follow through.

Overall, the paper is quite polished with a clear narrative. The sections on methods choices are clearly written from a researcher with a depth of experience in physicochemical analyses of ochre using multiple methodological approaches. I would see an improvement of the paper with some additional discussion on the potential impacts they have shown on provenance assignments. I’ve made some suggestions for minor edits needed for clarification in the attached PDF. I think the necessary revisions are minor. I applaud Dayet on a valuable contribution to the literature on ochre research.

Author Response

Response to reviewer 1 comment

Point1: What I think could be addressed in a bit more detail is how the differences in the methods may or may not have impacted Dayet’s own previous interpretations. In their earlier papers on the Diepkloof ochre assemblage, the results suggested some variation in the ochre provisioning, with at least some of the assemblage attributed to non-local shale. Have the new PIXE data (which wasn’t included in the 2013 paper), or any new insights developed since then, compelled Dayet to reinterpret the earlier data or reconsider any provenance assignments?

RESPONSE TO THE REVIEWER:

As mentioned in the paper, the results published in the 2013 and 2016 papers on Diepkloof rock shelter do take into account the conclusions of this methodological work (Line 476-481). In this respect, the data presented do not allow new archaeological interpretations, they are all in agreement with the previous published works. The only exception to this are the PIXE data of two archaeological pieces (13716 and 13773). The PIXE data were not included in the published provenance results. We did so because we had already observed that the PIXE analyses were much less sensitive and less accurate than ICP-MS data. We decided to publish all the PIXE data in a separate paper aiming to discuss the relevance of this method for provenance researches. Indeed, we have PIXE data for more archaeological pieces than those presented here (which are those analysed by ICP-MS and/or prepared for cross-sections only). The present work is the first step before we publish all the PIXE data we collected.

We propose to change the main text of the manuscript as follows:

Methodology

Line 476: “A final step consisted in selected a small but representative corpus of 28 archaeological ochre pieces for invasive analyses in order to evaluate the influence of post-depositional alterations on the composition of their surface and to confirm the relevancy of the results previously obtained. These methodological steps are described in a Ph.D. thesis (Dayet 2012). XRD and SEM-EDS methodological investigations were taken in consideration in all previous published articles, although they were never properly described (Dayet et al. 2013b; 2016; 2017). We will present here a summary of this research. The PIXE results were never used for archaeological application. The potential of the method will be discussed and we will use it to characterise post-depositional as well.”

Discussion

Line 805

“Finally, we showed that the contents of 6 trace elements are not affected by elemental migrations at the surface of archaeological samples, among which 5 are accurately dosed on the surface of geological samples in comparison with homogenized powders: As, Y, Nb, Pb and Th. This list is quite different from the list of trace elements dosed by ICP-MS that were used to discriminate the geological shale samples, which were As, Ba, Cr, Sb, and V (Dayet et al. 2016). Moreover, it is quite a low number of trace elements for the discrimination of geological ochre sources in general (Popleka-Filcoff et al. 2007; 2011; Eiselt et al. 2011; MacDonald et al. 2011; 2018; Attard Montalto et al. 2012; Domingo et al. 2012; Zipkin et al. 2015; 2020; Zarzycka et al. 2019; Pierce et al. 2020). Nonetheless, the discrimination between two sources on the basis of a few trace elements is possible as soon as these sources are characterised by consistent and significant differences. This might be the case, for instance, if they come from two different types of geological formations, as shown by a previous PIXE study (Mathis et al. 2014). It is also possible to carry out multi-variate analyses on this set of variables in order make efficient selection of the trace elements that have the highest weights in inter-source variations. After a careful consideration of the limitations of PIXE analyses done at the surface of heterogeneous ochre pieces (low representativeness of small area PIXE surface analyses for instance), and evaluation of post-depositional aspects, we can securely conclude that provenance researches by using non-invasive PIXE analyses are theoretically possible on this particular collection. Further investigations will be carried out on a larger selection of pieces in the future.

Point2: If any new assessments (or, even a positive reinforcement of the previous interpretations) have come to light with the new data, this would be a key point of discussion to interested readers. This is mentioned briefly in the conclusion, but mostly from a methodological standpoint and not an interpretive one. I think a stronger link to the earlier hypotheses posed in the previous publication could help drive the point home: how, in this case study, could the use of different methods impact the interpretation of ochre provisioning activities at Diepkloof? If the data patterns might have been interpreted differently in light of new or different information, this is important to know. If the data patterns are consistent and reinforce previous inferences, that would be equally important to highlight. If I had to suggest any additional content to add to the manuscript, a clearer and more detailed link to the previously published data that highlights any new interpretations would be a highly valuable follow through.

RESPONSE TO THE REVIEWER:

We do agree with the reviewer that we did not go as far as we could have in the critical evaluation of PIXE as a relevant method for addressing provenance issue for this particular collection. However, the aim of this paper is not to discuss this point. The aim of this paper is to discuss to which extend non-invasive methods can be used to address provenance researches. The case study is not what matters the most. We used the PIXE data of this particular collection to show that PIXE analyses can be used to address provenance issues after a careful consideration of instrumental limitations (low representativeness of small area PIXE surface analyses for instance), and post-depositional aspects (migration of elements at the surface of archaeological samples). What is important is not whether, at the end, we did find new attributions of provenance, but how we proceed to determine whether applying this final research step was possible or not. We showed that yes, it is possible (see the revision proposed above). We reached our goal. The archaeological implications of all PIXE data we have will be published in a separate paper.

Point3: I’ve made some suggestions for minor edits needed for clarification in the attached PDF. I think the necessary revisions are minor. I applaud Dayet on a valuable contribution to the literature on ochre research.

RESPONSE TO THE REVIEWER:

I did not find the pdf uploaded by the reviewer.

I am grateful to this reviewer for his/her precious comment and for helping me to improve the manuscript.

Reviewer 2 Report

The paper by Dayed is a good review of the published methods about Ochre and Iron-based raw materials used as pigments, and an interesting guideline for a multi-techniques characterization of such materials in the context of sites on the West coast of South Africa. The paper is complete and well written, only a few points should be taken into account. The title claims that the paper presents investigations on iron based “pigments”. In my opinion, using the term pigment is not completely correct, as the data presented refers in large part to raw minerals used as pigments. I’m wondering if the author could find a slightly more precise way to summarize the issue. Second point, I’m not sure that the term “taphonomic” used throughout the text to indicate alteration effects is completely correct.

Page 11: “Moreover, matrix effects are very strong in XRF analyses, making their calibration complex and time consuming when the samples analysed have different mineralogical compositions.” When making XRF calculations, calibration is seldom used because it very complicated. Usually comparison with known samples are preferred. I would change the sentence “matrix effects are very strong in XRF analyses, making the quantification of elements complex and time consuming”

Page 12, second paragraph: please change the sentence “macroscopic criteria in order to look for criteria”

Author Response

Response to reviewer 2 comment

Point 1: The title claims that the paper presents investigations on iron based “pigments”. In my opinion, using the term pigment is not completely correct, as the data presented refers in large part to raw minerals used as pigments. I’m wondering if the author could find a slightly more precise way to summarize the issue.

RESPONSE TO THE REVIEWER:

We agree with the reviewer. We had to let apart ochre use in paintings because it would have make the paper too long and would have decrease the balance between the literature review and the case study presented. We propose to change the title by using the comment of the reviewer:

“A methodological investigation of invasive and non-invasive analyses of ochre and iron-based pigment raw materials”

Second point, I’m not sure that the term “taphonomic” used throughout the text to indicate alteration effects is completely correct.

RESPONSE TO THE REVIEWER:

Using the term “taphonomy” or not is also a point we found tricky and difficult to resolve. Strictly speaking, what we deal with is indeed alteration effects. However, alteration effects are not restricted to archaeological contexts. Several geological phenomena can cause the alteration of a rock piece; all iron oxide forms can be submitted to alteration effects before being gathered by humans, especially weathering and secondary alterations. In order to keep this difference, we propose to use the more neutral adjective ‘post-depositional’ instead of ‘taphonomic’ all along the manuscript and to remove the term ‘taphonomy’ from the paper.

- Abstract, line 21: “We will then present a methodological approach that aimed to identify the instrumental limits and the post-depositional alterations that can skew the analytical results in the study of cohesive ochre fragments from the Diepkloof rock Shelter, South Africa.”

- Line 25 “We conclude that: non-invasive SEM-EDS and PIXE analyses provide non-representative results when the number of measurements is too low; post-depositional alterations caused significant changes in the mineralogical and major element composition at the surface of the archaeological pieces.”

- Key words: “Post-depositional alterations”

- Introduction Line 99 “We will characterise both instrumental limits and alteration effects due to post-depositional processes.

- Methodology Line 475 “A final step consisted in selected a small but representative corpus of 28 archaeological ochre pieces for invasive analyses in order to evaluate the influence of post-depositional alterations on the composition of their surface and to confirm the relevancy of the results previously obtained”

- Discussion Line 740 “3.3.2 Post-depositional alterations”

- Line 741 “The first post-depositional surficial modification identified from our results was the presence of deposits on the surface of several samples, despite the fact that this surface has been washed.”

- Line 745 “Two other patterns in our results can also be related to post-depositional alterations.”

- Line 759 “As regard to archaeological ferricrete pieces, post-depositional processes likely induced the migration of part of the elements at the surface sub-primary cortex, as was the case for most archaeological pieces.”

- Line 763 “Surprisingly, trace elements seem to be less affected than major elements by alteration processes.”

- Line 770 “Additionally, post-depositional processes likely contributed to the differences in iron oxide and oxi-hydroxide identified on the surface of archaeological pieces relative to powders.”

- Line 786 Having now determined both the potential instrumental limits and post-depositional alterations, we can now move forwards in the construction of a rigorous interpretive framework for our results.”

- Line 791 “However, this particular collection may represent an isolated case study given the salty nature of the superficial coatings of post-depositional origin identified on the archaeological pieces and the absence of similar compounds in their internal composition.”

- Line 828 “The methodology we followed, which was based on the comparison of invasive and non-invasive analyses of both geological and archaeological samples, was particular useful for demonstrating the respective limitations of these methods and the potential effects of post-depositional processes on the surfaces of archaeological specimens.”

Page 11: “Moreover, matrix effects are very strong in XRF analyses, making their calibration complex and time consuming when the samples analysed have different mineralogical compositions.” When making XRF calculations, calibration is seldom used because it very complicated. Usually comparison with known samples are preferred. I would change the sentence “matrix effects are very strong in XRF analyses, making the quantification of elements complex and time consuming”

RESPONSE TO THE REVIEWER:

We agree with the reviewer that a change of the sentence is necessary here.

We propose to change it as follow: “Moreover, matrix effects are very strong in XRF analyses, making the quantification of elements complex and time consuming when the samples analysed have very different mineralogical matrixes (see e.g. Mantler et al. 2006; Speakman and Shacley 2013).”

Page 12, second paragraph: please change the sentence “macroscopic criteria in order to look for criteria”

RESPONSE TO THE REVIEWER:

Changed: “macroscopic features in order to look for criteria”

I am grateful to this reviewer for his/her precious comment and for helping me to improve the manuscript.

Reviewer 3 Report

This paper deals with the characterisation of iron-based pigment in term of invasive and non-invasive analyses.

It seems a promising manuscript from the title, but the content lacks accuracy since there are statements that evidence author’ poor knowledge of analytical methods.

First, there is a serious problem with the definition of “non-invasive” analyses. Non-invasive requires an unambiguous definition. Nowadays, in the cultural heritage field, non-invasive analysis are more and more widespread, in order to interact as less as possible with the art objects. In fact, the growing demand of the use of instruments directly transportable in situ, is accompanied with the increasing demand of their minimum impact on the constituent materials. First, to better understand the importance in the choice of the most appropriate analysis for a particular art object is useful to distinguish between two categories of analyses: destructive/non-destructive and invasive/non-invasive. Destructive analysis are diagnostic methods that do not preserve the structural integrity and functionality of the art material, and generally, they require sampling. Non-invasive analysis, instead, do not alter the physical and the chemical conditions of the manufacture, allowing to not interfere with the changing processes between the manufacture and the environmental.

It seems that the author does not know the difference between non-invasive and non-destructive and in fact, she already writes in the abstract “We used ochre materials recuperated in both archaeological and geological contexts, and we compared non-invasive surface analyses by XRD, SEM-EDXS, and PIXE with analysis of powder pellets and sections from the same samples. We conclude that: non-invasive SEM-EDS and PIXE analyses”. On these premises, the author then built the whole article in a very wrong way.

I add that it is also strange to define the analyses as SEM-EDS and PIXE “surface analyses”.

The first 18 pages the author reviews a lot of experimental methods for the analysis of ochres, without going into details and describing the analytical techniques in a superficial way.

The second part of paper describes a case study that has been already published: the author herself states “They were taken in consideration in previous published works (Dayet et al. 2013b; Dayet et al. 2016) and we will now present a summary of the main results obtained”.

Because all of this, I recommend rejecting this paper.

Author Response

Response to reviewer 3 comment

First, there is a serious problem with the definition of “non-invasive” analyses. Non-invasive requires an unambiguous definition. Nowadays, in the cultural heritage field, non-invasive analysis are more and more widespread, in order to interact as less as possible with the art objects. In fact, the growing demand of the use of instruments directly transportable in situ, is accompanied with the increasing demand of their minimum impact on the constituent materials. First, to better understand the importance in the choice of the most appropriate analysis for a particular art object is useful to distinguish between two categories of analyses: destructive/non-destructive and invasive/non-invasive. Destructive analysis are diagnostic methods that do not preserve the structural integrity and functionality of the art material, and generally, they require sampling. Non-invasive analysis, instead, do not alter the physical and the chemical conditions of the manufacture, allowing to not interfere with the changing processes between the manufacture and the environmental.

It seems that the author does not know the difference between non-invasive and non-destructive and in fact, she already writes in the abstract “We used ochre materials recuperated in both archaeological and geological contexts, and we compared non-invasive surface analyses by XRD, SEM-EDXS, and PIXE with analysis of powder pellets and sections from the same samples. We conclude that: non-invasive SEM-EDS and PIXE analyses”. On these premises, the author then built the whole article in a very wrong way.

RESPONSE TO THE REVIEWER:

I am in disagreement with this reviewer. The rigorous definition of ‘non-destructive analysis’ is that the sample, whatever it is a sample of an archaeological, art object, or something else, is not destroyed during the analysis and thus available in future. This doesn’t imply that the archaeological or art object is kept intact. It is kept intact in case this non-destructive analysis is non-invasive too, which means, no matter/sample is removed from the object for the analysis. That’s why, in the abstract and all along the paper, I used the term ‘invasive’ when a part of the matter constituting the archaeological object is sampled; I used the expression ‘mimimally invasive’ when only micro-samples are taken or when a hole of micrometric diameter is made for the analysis, and I used both ‘surface analyses’ and ‘non-invasive analyses’ when no damage at all is made to the archaeological object, namely when no matter/sample is removed from it. This is the conventional way to use these terms in all analytical chemistry, except, perhaps, in cultural heritage studies. But Minerals is not a journal dealing with cultural heritage studies, and the aim of the paper is not to help with the conservation of archaeological samples. I do not think I should use the definition of ‘non-destructive’ proposed by the reviewer, because it is not commonly used, especially by the potential readers of the paper, mineralogists, geochemists or chemists, that will be familiar with the rigorous definition I use. Nonetheless, I acknowledge that the complementary aspect of ‘non-destructive’ and ‘non-invasive’ analyses is a bit confusing for someone who would not be familiar with it. I can propose to add a short note to clarify this point from the beginning of the literature review (part 2)

Line 171 “Choosing between the invasive and non-invasive mode for ochre analysis is more delicate than just choosing a method among a list. As a recall, ‘invasive’ means that a bit of matter is removed from the object or the sample analysed, while ‘non-invasive’ means that the object or the sample is kept intact (totally non-destructive).”

I add that it is also strange to define the analyses as SEM-EDS and PIXE “surface analyses”.

RESPONSE TO THE REVIEWER:

These methods are not generally speaking ‘surface analysis’, of course not. But I do qualify them as though when they are carried out on the surface of archaeological objects. None of the other reviewers had problems with this. I think that the revision we propose above make this point clear for the reader.

The first 18 pages the author reviews a lot of experimental methods for the analysis of ochres, without going into details and describing the analytical techniques in a superficial way.

RESPONSE TO THE REVIEWER:

The aim of this paper is not to describe all these methods. The reader is welcome to read the books and scientific articles I refer to if he/she wants to know more about each mentioned method. This part of the paper shows that ochre studies require a good knowledge of each method. The acquisition of this knowledge is not as simple as reading a single scientific article. Doing interdisciplinary research like archaeological ochre characterisation requires a complete training and specialization. There is no scientific article that can replace this. That’s why I decided not to describe each method, even in a short way. I decided to only recall the main factors that influence the accuracy and precision of each method in order to help researchers that already have a good knowledge of one of several of these methods to decide if it is worth to use them for addressing the archaeological question they are facing.

The second part of paper describes a case study that has been already published: the author herself states “They were taken in consideration in previous published works (Dayet et al. 2013b; Dayet et al. 2016) and we will now present a summary of the main results obtained”.

RESPONSE TO THE REVIEWER:

What is already published are the archaeological results that this methodological investigation allows to present. None of SEM-EDS and PIXE raw data presented in this case study were published before. We only presented broadly interpreted data of the SEM-EDS surface analyses of the archaeological samples in Dayet et al. 2013, not the raw data. We did present the powder XRD analyses of archaeological and geological samples in Dayet et al. 2016, but never the XRD analyses of the surfaces of geological samples. All the figures and the conclusions presented in this paper are new and entirely different from previous figures and results we published elsewhere.

The comments of this reviewer raises questions, but it seems to me that they are mostly misunderstandings of the scope and the fields of research this manuscript was wrote for.

Reviewer 4 Report

Firstly this paper reviews ochre studies by focusing on the analytical methods employed, the limits of non-invasive methods, as well as examples of some quality research addressing specific issues such as raw material selection and provenience, heat treatment. Then, it presents a methodological approach that aimed to identify the instrumental limits and taphonomic biases in the study of cohesive ochre fragments from the Diepkloof rock Shelter. XRD, SEM-EDXS, and PIXE results are presented and discussed.

It is a good paper, written with care, details and rigor. Even so, I have some comments to be considered by the author.

[1] Item 2.2

About the penetration depth of X-ray beams in XRF instruments, it was not taken into account considerations about the tube anode and applied voltage, as the penetration will not depend only on the effective atomic number of the matrix, but also on the energy profile of the X-ray tube beam, depending on the anode material and the applied voltage; these parameters can be varied to optimize the results in each situation.

It was not not discussed that the penetration of PIXE is much less than that of XRF, in addition to the difference in the size of the spots used, making the XRF measurement more interesting and representative, except when only the surface layer is targeted, in this case the PIXE is more interesting. 

[2] Table 5 - Symbols ( +, ++, ++, /, - , X) should be explained at the table legend or as bottom note.

[3] Legend of Table 8 - The information that trace elements are presented in ppm is missing at the legend.

[4] Figure 6 - A - “Diamonds in dark grey: edges of the section” are not shown, only Diamonds in black.

[5] AT the end of item 3 - Digitation correction:
The differences observed between the inside and the surface or the analysed samples favour incidental burning rather than intentional heating. >>>> change "or" by "of".

[6] Supplementary materials present results of SEM-EDXS and PIXE versus ICP, but nothing about these ICP data is cited or discussed in the item 3 of the paper text.

Author Response

Point [1] Item 2.2

About the penetration depth of X-ray beams in XRF instruments, it was not taken into account considerations about the tube anode and applied voltage, as the penetration will not depend only on the effective atomic number of the matrix, but also on the energy profile of the X-ray tube beam, depending on the anode material and the applied voltage; these parameters can be varied to optimize the results in each situation.

RESPONSE TO THE REVIEWER:

Yes, the reviewer is absolutely right, this is an essential information that I forgot or accidently erased when I wrote the manuscript. I am going to add it.

Line 211 “The penetration depth of X-ray beams in XRF instruments depends on the atomic number of the analysed material; the anode of the X-ray tube and the voltage applied (Potts et al. 1997b).”

It was not not discussed that the penetration of PIXE is much less than that of XRF, in addition to the difference in the size of the spots used, making the XRF measurement more interesting and representative, except when only the surface layer is targeted, in this case the PIXE is more interesting.

RESPONSE TO THE REVIEWER:

We propose to improve this part of the manuscript as follow:

Line 209 “SEM-EDXS and PIXE are designed for the analysis of small surfaces at a micrometric scale (not more than 1 mm2; Pollard et al. 2006; Pollard and Heron 2008; Kanngieser and Haschke 2006). They can be used for elementary analyses of thin layers because of the low critical penetration depth of secondary X-ray photons produced by electron or proton primary beams (Calligaro et al. 2004; Brisset et Boivin 2008; Beck et al. 2011).

Line 2012 “The spatial resolution of XRF analyses, except for micro-XRF methods, is higher than the one of SEM-EDX and PIXE analyses, as the XRF beam is higher than 1 mm in diameter and usually of several millimetres for portable instruments (Pollard et al. 2006; Pollard and Heron 2008; Kanngieser and Haschke 2006). This higher beam size is an advantage for the analysis of large areas and to do global analyses. Nonetheless, a high number of measurements is recommended for the analysis of heterogeneous rocks to be precise (Potts and West 2008).

[2] Table 5 - Symbols ( +, ++, ++, /, - , X) should be explained at the table legend or as bottom note.

RESPONSE TO THE REVIEWER:

Added.

“Legend : +++ : Main phase; ++ : secondary major phases (>10%) ; + : minor phases (<10%) ; ‐ : a single small peak.”

[3] Legend of Table 8 - The information that trace elements are presented in ppm is missing at the legend.

RESPONSE TO THE REVIEWER:

Added in the table.

[4] Figure 6 - A - “Diamonds in dark grey: edges of the section” are not shown, only Diamonds in black.

RESPONSE TO THE REVIEWER:

Removed from the legend, this was a mistake in the legend.

[5] AT the end of item 3 - Digitation correction:
The differences observed between the inside and the surface or the analysed samples favour incidental burning rather than intentional heating. >>>> change "or" by "of".

RESPONSE TO THE REVIEWER:

Changed.

[6] Supplementary materials present results of SEM-EDXS and PIXE versus ICP, but nothing about these ICP data is cited or discussed in the item 3 of the paper text.

RESPONSE TO THE REVIEWER:

Yes, they are. Table S1 is cited Line 562 and 587, in the Method section. They give an idea of the accuracy of PIXE and SEM-EDXS data of the instruments used, in the conditions of analysis, if we consider ICP-MS as reference values. This is only briefly mentioned in the method part given that this is not the main goal of the paper.

Many thanks to this reviewer for its very accurate and fine review that greatly helped to improve the paper.

Round 2

Reviewer 3 Report

Dear Dr Dayet,

yes we disagree on definition of "non-destructive" and /or " non invasive" analyses. In my opinion, and not only mine, depending on the information required, one might use a combination of truly non-invasive techniques (i.e. those which do not require a sample to be removed from the object, and which leave the object in essentially the same state before and after analysis), micro-destructive techniques (i.e. those which consume or damage a few picoliters of material and which may require the removal of a sample) and non-destructive techniques (i.e. a sample or complete object can be re-analyzed (with another technique) for further examination). The distinction between these techniques and types of analyses is of particular importance in the conservation field.

The new added sentence is a good compromise between our opinions.

Moreover at line 21 of abstract the sentence " we compared non -invasive surface analyses by XRF, SEM-EDS..." must be substitute by "we compared non -destructive analyses by XRF, SEM-EDS..."

I appreciate the effort to make in evidence the published data and the unpublished data.

The paper in this revised form is improved, and it can be published.